# Rethinking and Improving Robustness of Convolutional Neural Networks: a Shapley Value-based Approach in Frequency Domain

**Yiting Chen, Qibing Ren, Junchi Yan**[*]
Department of Computer Science and Engineering
MoE Key Lab of Artificial Intelligence
Shanghai Jiao Tong University
{sjtucyt, renqibing, yanjunchi}@sjtu.edu.cn

## Abstract

The existence of adversarial examples poses concerns for the robustness of convolutional neural networks (CNN), for which a popular hypothesis is about the frequency bias phenomenon: CNNs rely more on high-frequency components (HFC) for classification than humans, which causes the brittleness of CNNs. However, most previous works manually select and roughly divide the image frequency spectrum and conduct qualitative analysis. In this work, we introduce Shapley value, a metric of cooperative game theory, into the frequency domain and propose to quantify the positive (negative) impact of every frequency component of data on CNNs. Based on the Shapley value, we quantify the impact in a fine-grained way and show intriguing instance disparity. Statistically, we investigate adversarial training(AT) and the adversarial attack in the frequency domain. The observations motivate us to perform an in-depth analysis and lead to multiple novel hypotheses about i) the cause of adversarial robustness of the AT model; ii) the fairness problem of AT between different classes in the same dataset; iii) the attack bias on different frequency components. Finally, we propose a Shapley-value guided data augmentation technique for improving the robustness. Experimental results on image classification benchmarks show its effectiveness. The code for this paper is at `https://github.com/Ytchen981/CSA`

## 1  Introduction

Though convolutional neural networks (CNNs) have shown their great generalization power in various vision tasks, the existence of adversarial examples [37, 16, 24] show that CNNs are prone to be affected by adversarial noises that are imperceptible to humans. As inspecting the impact of frequency components, *i.e.* signals of a certain frequency in the frequency domain, one popular hypothesis about the brittleness of CNNs is that CNNs can exploit high frequency components (HFCs) that are not perceivable to humans [41, 42, 1, 21]. Based on this hypothesis, a line of works propose to utilize HFC filtering to improve the attack or defense methods [13, 22, 51]. However, towards the above *frequency bias* phenomenon, most works [41, 42, 21] intuitively divide the image frequency spectrum into two parts as low frequency components (LFCs) and HFCs then evaluate the contribution of each part based on the performance of models on the dataset, overlooking the possible disparity in sample-level and frequent component-level. Moreover, current works [17, 32] reveal that some successful adversarial attacks also depend on LFCs, which reveals the limitation of previous works: i) overlooking the contribution of each individual frequency component in LFCs and HFCs; ii) only do qualitative dataset-level analysis.

---

[*]Junchi Yan is the correspondence author, who is also with Shanghai AI Laboratory.

36th Conference on Neural Information Processing Systems (NeurIPS 2022).

Instead of studying the impact of HFCs and LFCs at the overall dataset level, in this paper we resort to quantify the contribution of each frequency component with the help of Shapley value [31]. Specifically, Shapley value is a metric from cooperative game theory which has been widely used in explainable machine learning [9, 2, 23, 36]. Since these works measure pixel-level importance for prediction in the spatial domain, our work is the first to apply Shapley value to the frequency domain. Shapley value satisfies several desirable properties for quantifying the contribution of each frequency component, and we justify that applying Shapley value in the frequency domain rather than spatial domain is a more natural approach on CNNs (c.f. Remark 1).

Based on the Shapley value, we quantify the contribution of each frequency component by measuring the average expected difference between the model's output brought by the absence of the frequency component regarding all possible circumstances where other frequency components are either masked or not. An illustration of the quantifying method is presented in Fig. 1. Through our quantification, one is able to inspect the contribution of each frequency components of each sample (instead of at dataset level). For simplicity, we refer to the frequency components with positive contribution as positive frequency components (PFC) and frequency components with negative contribution as negative frequency components (NFC).

Our quantification results provide insights in various perspectives. At the instance level, we present intriguing disparity between different data samples, *e.g.* model may rely more on HFCs to inference on some instances, as shown in the first row of Fig. 2, or rely more on LFCs to inference on the other instances, as shown in the second row of Fig. 2. Statistically, we inspect and analyze the adversarial training including the cause of the adversarial robustness achieved by adversarial training and the fairness problem in adversarial training (AT), which quantitatively proves that HFCs plays an important role in adversarial training. We further inspect and analyze the adversarial attack. We present the attack bias on different frequency components and provide a possible explanation for adversarial attacks focusing on LFCs [17, 32].

Based on our quantification, we propose a simple yet principled data augmentation method named Class-wise Shapley value-guided Augmentation (CSA) to correct the misalignment of features between different classes. By augmenting the data sample with features composed by NFCs of other data samples from the same class, the method boosts the adversarial robustness efficiently. Our major contributions are as follows:

• Instead of inspecting CNNs based on the manually divided frequency spectrum as done in existing literature [41], we introduce the Shapley value metric to the frequency domain and directly quantify the contribution of each individual frequency component on CNNs, with the advantage of desirable theoretical properties from the Shapely value itself (Section 3). To our knowledge, this is the first work to apply the Shapley value to the frequency domain on DNNs (specifically CNNs in this paper).

• As inspecting the impact of frequency components on models, we address the possible cause of adversarial robustness achieved by adversarial training and demonstrate that AT models improve robustness by sacrificing the utilization of HFCs(Sec. 4.2). We further address the fairness problem in adversarial training and demonstrate the negative relationship between the impact HFCs have on the ST model and the robust accuracy of the AT model(Sec. 4.3).

• We inspect the impact of different frequency components on adversarial attacks and demonstrate that adversarial attacks on PFCs and HFCs are relatively more effective than attacks on NFCs and LFCs. We further demonstrate that contribution of frequency components is not necessarily determined by its frequency and provide a possible explanation for successful adversarial attacks focusing on LFCs [17, 32](Sec. 4.4).

• Motivated by our explanatory results, we assume that NFCs compose features that are not correctly exploited by the model. Therefore we propose to augment data with NFCs found on data samples of its class, entitled by Class-wise Shapley value-guided Augmentation (CSA), which successfully boosts robustness in our empirical studies (Section 5).

## 2   Notations and Preliminaries

We present our model in the context of image classification, while we think the concept can still be applied to other backbones e.g. GNNs.

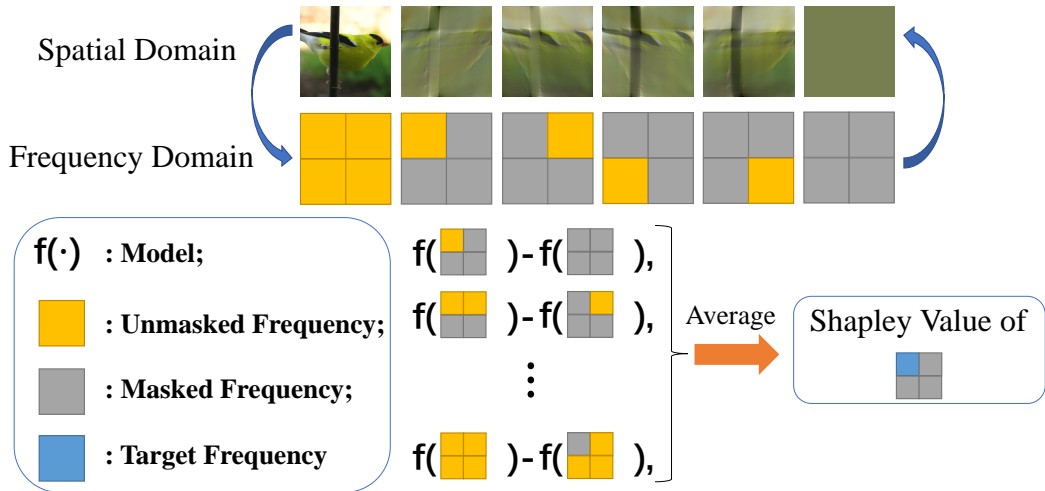

Figure 1: Illustration of the Shapley value calculated in frequency domain when the frequency components are devided into $2 \times 2$ patches. The Shapley value of a frequency component is the average marginal effect that the frequency component contributes to the output of the model.

**Notation:** $\langle \mathbf{X}, \mathbf{y} \rangle$ denotes a data sample where $\mathbf{X} \in \mathbb{R}^{d_1 \times d_2}$ is an image and $\mathbf{y} \in \{1, \ldots, C\}$ is its label where $C$ is the number of classes. For simplicity, we omit the channel of images. $f(\cdot, \theta) : \mathbb{R}^{d_1 \times d_2} \to \mathbb{R}^C$ denotes a neural network that takes image as input and output a prediction where $\theta$ represent its parameters. We use $\ell(\cdot, \cdot)$ to denote loss function and $\| \cdot \|_p$ to denote $\ell_p$ norm. $\mathcal{F}(\cdot) : \mathbb{R}^{d_1 \times d_2} \to \mathbb{C}^{d_1 \times d_2}$ denotes the Discrete Fourier Transfom (DFT). $\mathcal{F}^{-1}(\cdot) : \mathbb{C}^{d_1 \times d_2} \to \mathbb{R}^{d_1 \times d_2}$ denotes the inverse Discrete Fourier Transform.

**Discrete Fourier Transform (DFT).** DFT transforms finite signals into complex-valued functions of frequency. The 2D-Discrete (for image data considered in this paper) Fourier Transform is given by:

$$\mathcal{F}(\mathbf{X})(u, v) = \sum_{m=0}^{d_1-1} \sum_{n=0}^{d_2-1} \mathbf{X}(m, n) e^{-i2\pi(\frac{mu}{d_1} + \frac{nv}{d_2})} \quad (1)$$

The inversion of DFT is computed as:

$$\mathbf{X}(m, n) = \frac{1}{d_1 d_2} \sum_{u=0}^{d_1-1} \sum_{v=0}^{d_2-1} \mathcal{F}(\mathbf{X})(u, v) e^{i2\pi(\frac{mu}{d_1} + \frac{nv}{d_2})} \quad (2)$$

**Adversarial Attack.** It aims to find a small noise $\delta$ bounded in $\ell_p$ space to cause failure [37, 16].

$$\delta = \arg \max_{\delta} \ell(f(\mathbf{X} + \delta), \mathbf{y}), \quad s.t. \quad \|\delta\|_p \leq \epsilon \quad (3)$$

**Defense.** Adversarial training [24] becomes the most effective defense method [3], which is typically conducted by minimizing loss directly on adversarial training data.

$$\theta = \arg \min_{\theta} \mathbb{E}[\max_{\|\delta\|_p \leq \epsilon} \ell(f(\mathbf{X} + \delta), \mathbf{y})] \quad (4)$$

**Definition of the Shapley Value:** Let $\mathcal{N} = \{1, \cdots, n\}$ denotes the finite set of players. Each non-empty subset $\mathcal{S} \subseteq \mathcal{N}$ is called a coalition. A cooperative game is defined by the pair $(\mathcal{N}, V)$, where $V : 2^{\mathcal{N}} \to \mathbb{R}$ is a mapping that assigns a real number to each coalition and satisfies $V(\emptyset) = 0$. Let $\pi \in \Pi(\mathcal{N})$ denotes a permutation of player set $\mathcal{N}$ and $\pi(i)$ is the position of player $i \in \mathcal{N}$, where $\Pi(\mathcal{N})$ is the set of permutations. Then the predecessor set of player $i \in \mathcal{N}$ in permutation $\pi$ is $P_i^{\pi} = \{j \in \mathcal{N} | \pi(j) < \pi(i)\}$. The Shapley value $\phi_i^V$ of player $i$ is defined as:

$$\phi_i^{\mathcal{N}, V} = \frac{1}{|\Pi(\mathcal{N})|} \sum_{\pi \in \Pi(\mathcal{N})} [V(P_i^{\pi} \cup \{i\}) - V(P_i^{\pi})] \quad (5)$$

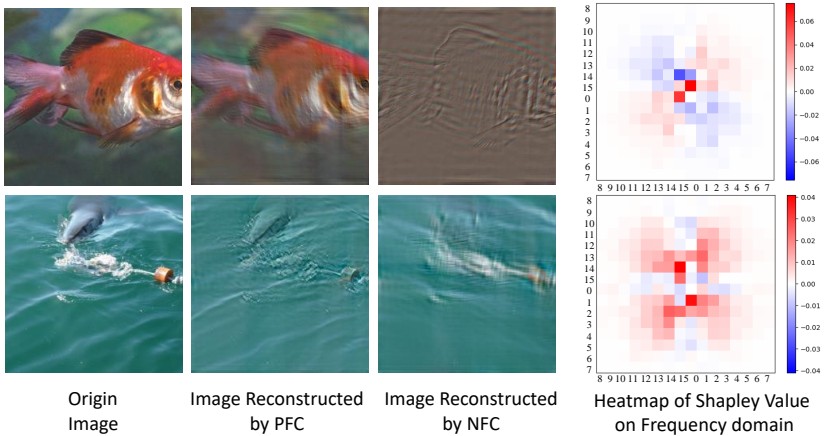

| Origin Image | Image Reconstructed by PFC | Image Reconstructed by NFC | Heatmap of Shapley Value on Frequency domain |

Figure 2: Shapley value based quantification of two example images in ImageNet. Left to right: origin image, image reconstructed with positive frequency components (PFC), images reconstructed with negative frequency components (NFC), the heat map of Shapley value in frequency domain. Note that the heatmap is shifted, the closer to the center the lower the frequency.

The Shapley value [31] of the player is the average marginal contribution of the player to the value of predecessor set in every possible permutation.

**Properties of Shapley Value:** It satisfies many desirable properties including *Null player*, *Efficiency*, *Symmetry*, *Linearity*. See Appendix D for more details about the properties of Shapley value.

## 3 Shapley Value-based Quantification

### 3.1 Contribution quantification for frequency component

We assume that frequency components in the data sample cooperate with each other and compose the feature that the model process to infer. To apply the Shapley value to the frequency components, we firstly construct a cooperative game in this scenario. Considering frequency components as players, we define the set of players for data sample $\langle \mathbf{X}, \mathbf{y} \rangle$ as:

$$\mathcal{N}_{\mathbf{X}} = \{\mathcal{F}(\mathbf{X})(0,1), \cdots, \mathcal{F}(\mathbf{X})(d_1 - 1, d_2 - 1)\} \tag{6}$$

Note that $\mathcal{F}(\mathbf{X})(0,0)$ determines the mean value of the image, we exclude this frequency component from the player set and keep the mean value of the image unchanged to fairly calculate the marginal contribution of the frequency components.

For a model $f(\cdot)$, we define the output of the model on $S \subseteq \mathcal{N}$ as:

$$f(S) = f\left(\mathcal{F}^{-1}(\sigma_S \odot \mathcal{F}(\mathbf{X}))\right)_{\mathbf{y}} \tag{7}$$

where $f(\cdot)_{\mathbf{y}}$ denotes the $\mathbf{y}$-th output of the model and $\odot$ denotes Hadamard product. $\sigma_S \in [0,1]^{d_1 \times d_2}$ is the mask generated according to $S$ satisfying:

$$\sigma_S(u,v) = \mathbb{I}[\mathcal{F}(\mathbf{X})(u,v) \in S] \tag{8}$$

where $\mathbb{I}(\cdot)$ denotes an indicator that output 1 when the condition is satisfied and 0 otherwise. Specifically, $\sigma_S(0,0) = 1$. The corresponding characteristic function of the game is defined as:

$$V(S) = f(S) - f(\emptyset), S \in \mathcal{N} \tag{9}$$

By definition, we calculate the Shapley value $\phi_{u,v}^{\mathcal{N}_{\mathbf{X}},f}$ of frequency component $\mathcal{F}(\mathbf{X})(u,v)$ according to Eq. 5. The Shapley value of a frequency component evaluates the average marginal contribution of the frequency component to the neural network's output of the ground-truth class, which takes desirable properties including *Null player*, *Efficiency*, *Symmetry* and *Linearity*.

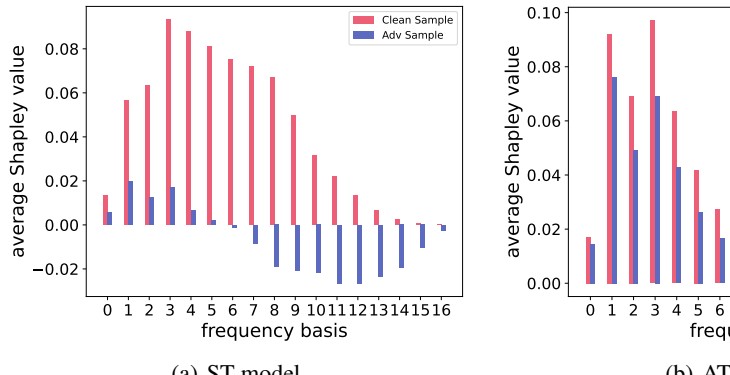

(a) ST model                         (b) AT model

Figure 3: Average shapley value of frequency componets in clean samples and adversarial samples over frequency basis of ResNet18 on CIFAR10.

## 3.2 Justification for frequency based Shapley value on CNNs

When it comes to measuring the contribution of input features in images, the common approach is decomposing features into pixels in the spatial domain and considering each pixel as a player [9, 2, 23, 36]. Missing players that are not in a coalition $S$ are replaced with a reference value. The reference value are often chosen intuitively as zero or the mean value of the image. The choice of reference value may affect the explanatory results on CNNs.

**Remark. 1 (Decomposition of convolution)** *The output of a convolution operation $f_{conv}(\cdot)$ on $\mathbf{X}$ can be decomposed into a linear function of output on frequency components on $\mathbf{X}$. Formally,*

$$f_{conv}(\mathbf{X}) = \sum_{u=0}^{d_1-1} \sum_{v=0}^{d_2-1} \left[ \mathcal{F}(\mathbf{X})(u,v) f_{conv} \left[ \mathcal{F}^{-1}(\mathbf{I}_{d_1 \times d_2}^{u,v}) \right] \right] \tag{10}$$

where $\mathbf{I}_{d_1 \times d_2}^{u,v} \in \mathbb{R}^{d_1 \times d_2}$ is a Notch pass filter *i.e.* it contains zeros except the element on the $u$-th row and $v$-th column is one. According to Remark 1, the output of a convolution operation is a linear function *w.r.t.* each frequency component. Hence the reference value for frequency components is naturally zero.

Furthermore, the Shapley value intuitively aligns more with what is important in an additive setting, as mentioned in previous works [20]. For non-additive models, Shapley value may fail to explain the importance of players. For example, function $f(x) = x_1 x_2 x_3$ with reference value set to be zero. The Shapley value of each variable is $\frac{1}{3} f(x)$ while the value of each variable may vary. For frequency components, the CNN is approximately a linear function. See Appendix D for more details.

## 4 Rethinking Robustness in Frequency Domain

In this section, we conduct extensive experiments to give a fundamental analysis of the contributions of different frequency components. In Sec. 4.1, we introduce details and results of instance-level sampling for Shapley value calculation. In Sec. 4.2, we inspect the contribution of frequency components of clean (adversarial) data with standard (adversarial) training paradigms respectively, followed by an in-depth analysis of the way how adversarial attacks work in Sec. 4.4. We demonstrate the discussion of fairness in Sec. 4.3

### 4.1 The Shapley Value of Frequency Components and The Instance Disparity

To get the Shapley value of each frequency component, we employ Monte Carlo Permutation Sampling to approximate the Shapley value [31]. We sample the Shapley value on CIFAR10, CIFAR100, Tiny ImageNet and ImageNet.

**Quantification result** We demonstrate our quantification result on ImageNet in Fig 2. Each row contains the results of a data sample. We show the original image, the reconstructed image from

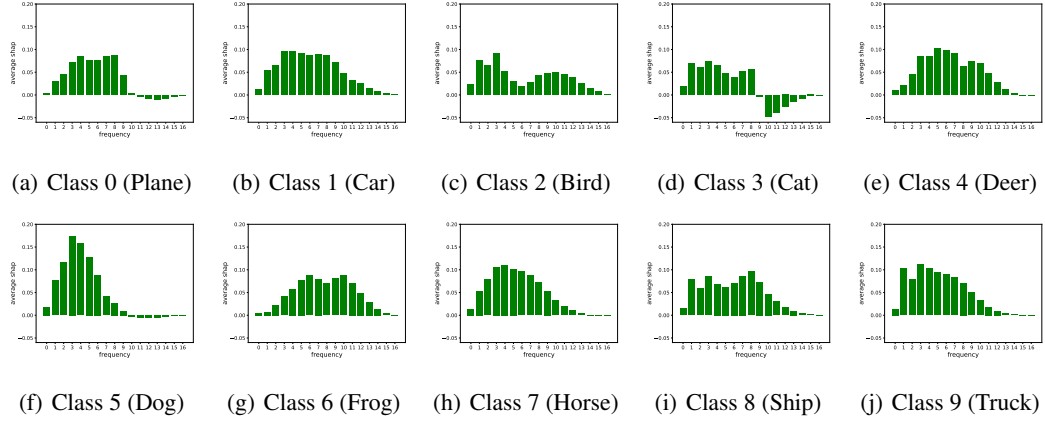

Figure 4: Average Shapley value over frequencies of clean samples on ST model from CIFAR10.

positive frequency components (PFCs), the reconstructed image from negative frequency components (NFCs), and the heatmap of Shapley value from left to right.

The Shapley value of each frequency component varies between different data samples. As shown in Fig. 2, the NFCs in the first image are mainly of high frequency or medium frequency while the PFCs are mostly low frequency. As a result, the NFCs compose the contour of the goldfish and other textures while PFCs recover the main body of the fish just like how humans perceive it. However, the "shark" image in the second row shows the opposite direction to "goldfish": PFCs mainly contains the contour of the shark, i.e., the HFC information, which means the *frequency bias* varies at the instance-level. See Appendix A for more details and more results.

## 4.2 The Cause of the Adversarial Robustness of AT model

Observations in Sec. 4.1 are based on clean examples. We further inspect the difference between ST and AT models in the frequency domain based on our Shapley value metric. For CIFAR10, adversarial examples are generated with PGD-20 with a Linf bound of $8\backslash255$ and step size of $2\backslash255$. We show results of ResNet-18 on CIFAR10. Refer to Appendix B for more results of vgg16 on CIFAR10 and ResNet-18 on CIFAR100 and refer to Appendix A for more details of the model training.

### 4.2.1 Standard Trained Models Mainly Exploit LFC but Are More Vulnerable on HFC

As shown in Fig. 3(a), first, for clean examples, LFCs have significant positive Shapley value on average for standard trained (ST) model while the average Shapley value of HFCs are nearly zero, which means ST model barely exploits HFC; Second, for adversarial examples, the mean Shapley value of HFC are significantly negative, which shows that ST model is more vulnerable on HFC.

### 4.2.2 Adversarial Trained Models Improve Robustness by Sacrificing Information in HFCs

As shown in Fig. 3(b), LFCs have high positive Shapley value on both clean and adversarial examples and the average Shapley value of HFCs are nearly zero on both clean and adversarial examples, which may be the key to improved robustness of AT. We hypothesize that AT tends to filter out the negative impact of HFC and the outcome is that AT focuses more on LFC, overlooking information in HFC. As Fig. 3(a) shows, the HFC of clean images plays a remarkable positive impact. Thus we conjecture that the trade-off between the clean accuracy and the robust accuracy of AT is possibly the result of whether the model utilizes the information in HFCs.

## 4.3 Fairness of AT in Frequency Domain

The class-wise disparity of robustness of AT model is a troublesome phenomenon even on the balanced dataset such as CIFAR-10 [43, 38] which results in fairness concerns. Though the clean accuracy of the ST model on each classes is similar, the robust accuracy of the AT model differs between classes. For CIFAR10, the class 0, 1, 8, 9 that belong to the same coarse class "Transportation" exhibit relatively high accuracy and robustness while other classes related to "Animals" are less robust except the class 7 "horse".

To investigate the difference between different classes, we demonstrate the average Shapley value of clean data samples from each class for ST model in Fig. 4. We generally find the trend that the absolute Shapley value of HFC is relatively larger on non-robust classes (*e.g.* class 2,3,4 and 5) than robust classes (*e.g.* class 1, 7, 8 and 9).

We further calculate the average absolute Shapley value of HFC (we intuitively take components with frequency higher than $70\%$ of the largest frequency) for ST model. As shown in Fig. 5, there is a negative relationship between the adversarial accuracy of AT model and the average absolute Shapley value of HFC for ST model (the Pearson correlation coefficient is $-0.8765$). As shown in Sec. 4.2, HFCs compose useful but vulnerable information while AT tends to sacrifice HFCs for robustness. Therefore, we conjecture that: *classes, where HFCs have a larger impact on the ST model, would have relatively lower adversarial accuracy under AT.*

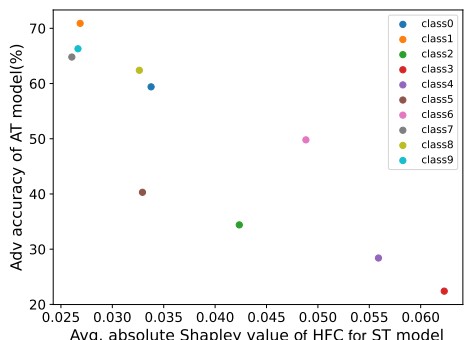

Figure 5: The negative relationship between the adversarial accuracy of AT model and the average absolute Shapley value of HFC for ST model.

### 4.4 Inspecting Adversarial Attack in Frequency Domain

#### 4.4.1 Adversarial Attacks on PFCs and HFCs Are More Effective

In this section, we inspect the performance of adversarial attacks over different frequency components. Specifically, for an adversarial noise $\delta$ generated by Eq. 3 on ST model $f$ and data sample $\langle \mathbf{X}, \mathbf{y} \rangle$, we transform the noise $\delta$ into frequency domain as $\mathcal{F}(\delta)$, which is further applied by masks based on different metrics to determine the impact of adversarial noise on different frequency components.

We first build masks based on the division between PFCs and NFCs of the data sample and the masked noise is defined as follows, where $\odot$ is Hadamard product:

$$
\begin{aligned}
\delta_{PFC} = \mathcal{F}^{-1}(\sigma_{PFC} \odot \mathcal{F}(\delta)), & \qquad \sigma_{PFC}(u, v) = \mathbb{I}(\phi_{u,v}^{X,f} > 0) \\
\delta_{NFC} = \mathcal{F}^{-1}(\sigma_{NFC} \odot \mathcal{F}(\delta)), & \qquad \sigma_{NFC}(u, v) = \mathbb{I}(\phi_{u,v}^{X,f} < 0)
\end{aligned}
\tag{11}
$$

Similar to Eq. 11, we define adversarial noises $\delta$ on HFCs and LFC as

$$
\begin{aligned}
\delta_{LFC} = \mathcal{F}^{-1}(\sigma_{LFC} \odot \mathcal{F}(\delta)), & \qquad \sigma_{LFC}(u, v) = \mathbb{I}[r(u, v) < \epsilon_{low} \cdot r_{max}] \\
\delta_{HFC} = \mathcal{F}^{-1}(\sigma_{HFC} \odot \mathcal{F}(\delta)), & \qquad \sigma_{HFC}(u, v) = \mathbb{I}[r(u, v) > \epsilon_{high} \cdot r_{max}]
\end{aligned}
\tag{12}
$$

where $r(u, v)$ is the frequency of the frequency component, $r_{max}$ is the maximum frequency in this data sample, $\epsilon_{low}$ and $\epsilon_{high}$ is the intuitively determined threshold for low frequency and high frequency. In our experiment, $\epsilon_{low} = 0.4$, $\epsilon_{high} = 0.6$.

We test the error rate of ST model as the input data sample is attacked with various noises, as shown in Fig. 6. The error rate is nearly zero for origin data sample $X$ and nearly one hundred percent for data samples attacked with the unmasked noises $X + \delta$. For adversarial noise on PFCs and NFCs, $X + \delta_{PFC}$ reaches a similar error rate of ST model as $X + \delta$ which is significantly larger than the error rate with $X + \delta_{NFC}$. It indicates that the adversarial attack on PFCs is more effective. Similar for HFCs and LFCs, the adversarial attack on HFCs is more effective than NFCs.

#### 4.4.2 A Possible Explanation for Adversarial Attacks Focusing on LFCs

As demonstrated in Sec. 4.2 and Sec. 4.3, we quantitatively proved that models are more vulnerable on HFCs than on LFCs. Previous works hypothesize that exploiting HFCs causes the brittleness of CNNs [41, 40, 45]. However, it conflicts with the success of the adversarial attacks that are limited on LFCs [17, 32] as addressed in [4, 25] and demands a closer look.

To explain such an intriguing conflict, we calculate the ratio of PFCs and NFCs for each frequency over the spectrum. As shown in Fig. 7, both PFCs and NFCs that have a significant impact take up a

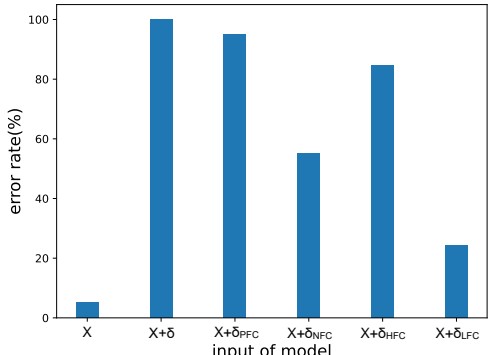

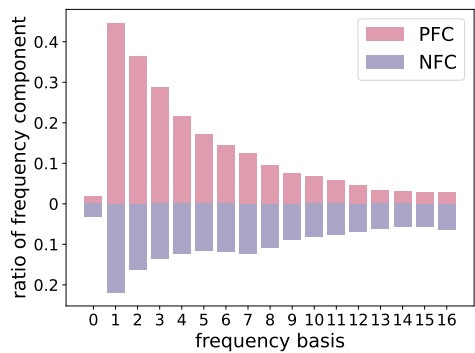

Figure 6: The error rate of ST model on CIFAR10 with different input. $X$ represent origin data. $\delta$ is the adversarial noise generated against the model. $\delta_{PFC}$, $\delta_{NFC}$, $\delta_{HFC}$ and $\delta_{LFC}$ are the masked adversarial noise that are constrained on PFC, NFC, HFC and LFC.

Figure 7: Ratio of the component PFC and NFC with significantly large Shapley value (absolute value greater than $20\%$ of the maximum absolute Shapley value of the same frequency). Results are averaged over clean examples on ST model (ResNet18) on CIFAR10.

fair ratio (at least $0.1$) over the whole frequency spectrum. Therefore, we argue that *the contribution of frequency components on the prediction of CNNs are not necessarily related to its frequency*.

The fact that PFCs and NFCs actually co-exist on every frequency basis provides a possible explanation for the success of the adversarial attacks that are limited on LFCs [17, 32]. Since the ratio of PFCs and NFCs are even higher at low frequency, it is possible for adversarial attacks that focus on LFCs to reduce the impact of PFCs and boost the impact of NFCs.

## 5 Improving Robustness via Class-wise Shapley Value-guided Augmentation

**Motivation.** As defined in Sec. 3, frequency components with negative Shapley value make the model output lean to other classes rather than the ground-truth class, which implies that the distribution of image features composed by NFC is mistakenly aligned with that of other classes. Intuitively, by proactively correcting the misalignment of features of the model, we could improve the adversarial robustness for future attacks. Moreover, our observations in Sec. 4.3 reveal that there exists obvious disparity at the class level of distribution of NFC. Therefore, we propose to actively augment data with features composed by NFC of the same class, named Class-wise Shapley value-guided Augmentation (CSA). To be specific, based on a CNN $f$, we obtain the CSA $p_X^f$ of data $X$ by masking its PFC as follows, with the indicator function $\mathbb{I}[\cdot]$:

$$p_X^f(u, v) = \mathbb{I}[\phi_{u,v}^{X,f} < 0] \cdot \mathcal{F}(X)(u, v) \tag{13}$$

**Implementation** Since it takes a large number of samplings for obtaining a stable Shapley value (c.f. Eq. 5) for each sample, it is necessary to downgrade the time complexity. Fortunately, we find that for

Table 1: Test accuracy of ResNet-18 trained on CIFAR10 and CIFAR100 with CSA. 'Final' denotes the performance at the last training epoch. 'Best' refers to the best snapshot model in training.

| **CIFAR10** | | Natural Accuracy w/o attack | | | attacked by PGD-20 | | | attacked by AutoAttack | | |
|---|---|---|---|---|---|---|---|---|---|---|
| | | Best | Final | Diff | Best | Final | Diff | Best | Final | Diff |
| PGD-AT [24] | baseline | **83.83%** | **84.49%** | -0.66 | 49.87% | 46.38% | 3.49 | 46.52% | 44.06% | 2.46 |
| | **CSA** | 81.57 % | 82.91 % | **-1.34** | **51.13%** | **49.42%** | **1.71** | **47.17%** | **46.56%** | **0.61** |
| TRADES [49] | baseline | **81.23%** | **81.71%** | -0.48 | 51.18% | 51.08% | **0.10** | 47.94% | 47.74% | 0.20 |
| | **CSA** | 80.84 % | 81.62 % | **-0.78** | **52.57%** | **52.06%** | 0.51 | **48.94%** | **49.16%** | **-0.22** |
| **CIFAR100** | | Best | Final | Diff | Best | Final | Diff | Best | Final | Diff |
| PGD-AT [24] | baseline | **56.55%** | **57.13%** | -0.58 | 24.38% | 21.70% | 2.68 | 21.66% | 20.10% | 1.56 |
| | **CSA** | 56.31 % | 56.72 % | **-0.41** | **26.14%** | **23.63%** | **2.51** | **23.02%** | **21.64%** | **1.38** |
| TRADES [49] | baseline | 53.09% | 53.09% | **0.00** | 27.01% | 27.01% | **0.00** | 22.76% | 22.76% | **0.00** |
| | **CSA** | **54.88%** | **54.46%** | 0.42 | **27.81%** | **27.77%** | 0.04 | **23.61%** | **23.73%** | -0.12 |

Table 2: Test accuracy of VGG16 and WRN-28-10 trained on CIFAR10 with CSA. 'Final' denotes the performance at the last training epoch. 'Best' refers to the best snapshot model in training.

| **WRN-28-10** | | Natural Accuracy w/o attack | | | attacked by PGD-20 | | | attacked by AutoAttack | | |
|---|---|---|---|---|---|---|---|---|---|---|
| | | Best | Final | Diff | Best | Final | Diff | Best | Final | Diff |
| PGD-AT [24] | baseline | **85.90%** | 85.67% | 0.23 | 50.18% | 47.51% | **2.67** | 47.86% | 45.69% | 2.17 |
| | **CSA** | 85.48 % | **85.87 %** | **-0.39** | **53.00%** | **50.83%** | 3.50 | **49.61%** | **48.75%** | **0.86** |
| TRADES [49] | baseline | 84.13% | 84.73% | **-0.60** | 53.17% | 48.72% | 4.45 | 50.21% | 46.65% | 3.56 |
| | **CSA** | **84.41%** | **84.92%** | -0.51 | **54.32%** | **50.53%** | **3.79** | **51.40%** | **48.65%** | **2.75** |
| **VGG16** | | Best | Final | Diff | Best | Final | Diff | Best | Final | Diff |
| PGD-AT [24] | baseline | **77.90%** | **80.06%** | -2.16 | 48.03% | 46.96% | 1.07 | **43.68%** | 42.70% | 0.98 |
| | **CSA** | 75.42 % | 75.40 % | **-0.02** | **48.22%** | **48.06%** | **0.16** | **43.68%** | **43.53%** | **0.15** |
| TRADES [49] | baseline | 79.86% | 79.69% | 0.17 | 46.92% | 46.63% | **0.29** | 42.50% | 42.67% | -0.17 |
| | **CSA** | **80.09%** | **80.23%** | **-0.14** | **48.58%** | **48.23%** | 0.35 | **44.13%** | **43.97%** | **0.16** |

Table 3: Testing accuracy of ResNet18 trained on CIFAR10 with high frequency suppressing methods. 'baseline' denotes the original TRADES method, 'sup HFC' denotes combining the suppressing high frequency method with TRADES and 'CSA' denotes combining our CSA with TRADES.

| Methods | | Natural Accuracy w/o attack | | | attacked by PGD-20 | | | attacked by AutoAttack | | |
|---|---|---|---|---|---|---|---|---|---|---|
| | | Best | Final | Diff | Best | Final | Diff | Best | Final | Diff |
| | baseline | **81.23%** | **81.71%** | -0.48 | 51.18% | 51.08% | **0.10** | 47.94% | 47.74% | 0.20 |
| TRADES [49] | sup HFC | 78.14% | 80.26% | -2.12 | 50.66% | 50.42% | 0.24 | 45.47% | 45.14% | 0.33 |
| | **CSA** | 80.84% | 81.62% | **-0.78** | **52.57%** | **52.06%** | 0.51 | **48.94%** | **49.16%** | **-0.22** |

the same class, the feature distribution composed by NFC aligns well between samples, and depending on only a small ratio of whole data (3% in our experiment) brings out an obvious improvement. Detailed analyses at a statistical level and experimental level can be found in Appendix B.

For each class $c$, we randomly take data samples $\langle X_1, c \rangle, \langle X_2, c \rangle, \cdots, \langle X_n, c \rangle$ and generate a set of CSAs with a pretrained model $\hat{f}$ as: $\mathcal{P}_c^{\hat{f}} = \{p_{X_1}^{\hat{f}}, p_{X_2}^{\hat{f}}, \cdots, p_{X_n}^{\hat{f}}\}$ and train our model as follows:

$$\theta = \arg\min_\theta \mathbb{E}\left[\max_{\delta < \|\epsilon\|_p} \ell\left(f[\mathcal{F}^{-1}(\mathcal{F}(X) + \alpha * p^{\hat{f}}) + \delta], y\right)\right], p^{\hat{f}} \in \mathcal{P}_y^{\hat{f}} \qquad (14)$$

where the $\alpha$ is a coefficient of the CSA $p^{\hat{f}}$.

**Results** We conduct experiments on CIFAR10 and CIFAR100 [19] to verify the effectiveness of our CSA. We train ResNet18 [18] on CIFAR10 and CIFAR100 under PGD-AT [24] and TRADES [49]. Attacks are crafted within $\ell_\infty$ bound with $\epsilon = 8/255$. We select PGD-20 attack [24] and AutoAttack (AA) [12] as the attack baselines. As Table 1 shows, under various attacks, our CSA consistently achieves better robustness over defense baselines with little cost of natural accuracy. With the Shapley value calculated with ResNet-18, we further apply our CSA on VGG16 [33] and WideResNet-28-10 [46] on CIFAR10. As shown in Table 2, our CSA also greatly improves the adversarial robustness of VGG16 [33] and WRN-28-10 [46]. It indicates that our CSA method is model-agnostic and could be applied to various model architectures. Note that the CSA set of each class contains the NFC of the first 150 images without shuffling for CIFAR10 and 30 images for CIFAR100, *i.e.*, 3% of the training set in CIFAR10 and 6% of the training set in CIFAR100, which shows that obtaining CSA only takes little overhead and the feasibility of our method in realistic settings. Training details and ablation study can be found in Appendix C.

**Comparison between CSA and high frequency filtering defense method** High frequency filtering methods have been proposed to defend adversarial attacks [13, 22, 51]. Most high frequency filtering methods are proposed to deal with $l_2$ norm adversarial attack and are compared with other non-adversarial training defend methods. We take the suppressing high frequency method proposed in [51] as baseline, which masks the high frequency components and trains the model with TRADES, and test the robust accuracy of ResNet18 on CIFAR10 with $\ell_\infty$ attack where $\epsilon = 8/255$.

As demonstrated in Tab. 3, for $l_\infty$ attack, suppressing high frequency components even slightly degrade the testing accuracy on CIFAR10.

# 6 Related Work

**Adversarial Attack** Since the seminal works [37, 16] have been proposed that reveal the vulnerability of DNNs, there is an increasing interest in studying stronger adversarial attacks, such as FGSM [16], C&W attack [5], PGD attack [24], and AutoAttack [12]. The robustness of CNNs in this work has been thoroughly evaluated on these above attack baselines.

**Adversarial Defense** The development between attack and defense is just like an arms race: the existence of adversarial examples promotes the development of defense methods, among which adversarial training (AT) has been considered the most effective one [3]. However, AT has its own limitations: i) the trade-off between accuracy and robustness [34, 39, 48, 44], ii) *robust overfitting* [30, 10], which brings a large robustness generalization gap. Many works have been proposed to improve AT by sample-wise importance re-weighting [50], adding more unlabeled data [27, 6] or regularizing the embedding space [29]. A recent work [28] proposes to augment data with the approximated confounders from a causal perspective for improved robustness while these confounders are class-agnostic and lack theoretical guarantee. In this work, we quantify the frequency importance via the theoretically stable Shapley value and design a class-specific data augmentation strategy, which achieves great improvement in robustness with better robustness generalization.

**Understanding Robustness in Frequency Domain** One way to inspect the intriguing nature of adversarial attack and defense is through the lens of frequency spectrum. Many works explore the sensitivity of CNNs and frequency properties of the model [40, 45, 1] and study whether convolution itself has an intrinsic bias in frequency domain [7]. A hypothesis is that CNNs exploit HFCs which leads to the lack of robustness [41]. Driven by this hypothesis, many pre-processing defend methods have been proposed [13, 22, 51]. Meanwhile, some works also come up with adversarial attacks focusing on LFC [17, 32] which leads to assumptions that constituent frequencies of adversarial examples are dependent on the dataset [4, 25]. Efforts to quantify and inspect the contribution of each frequency component have been made [42, 21]. We employ Shapley value, a tool that has preferable properties, to quantify the contribution of each frequency component on the fine-grained level.

**Explainability with Shapley Value** Shapley Value [31] is proposed to distribute contributions among players in a game. It has been widely applied for the explainability of deep learning. Some works propose its approximations [23, 9, 2]. While another line of works focus on the condition to apply Shapley value [14, 15]. Also, the interaction between input features [36, 35, 47] is recently studied. In our work, we apply Shapley value in the frequency domain to quantify the contribution of frequency components. The quantification have many desirable theoretical properties.

# 7 Conclusion and Outlook

We have made efforts to quantify the contribution of each frequency component on CNNs. We employ Shapley value, a method with theoretically desirable properties, to quantify the contribution of a single frequency component on the instance-level. Through our quantification, we explore the difference between standard trained (ST) model and adversarial trained (AT) model. Our analysis of the fairness in AT provides insights of the cause of robustness difference over classes. We further examine the attack effect of adversarial noises on different frequency components. Based on our findings, we then propose a simple yet effective class-wise data augmentation method CSA which augments data with the NFC found in this class. Experimental results have shown its effectiveness.

**Limitation & future work:** Our current results are limited mainly to CNNs while we believe it is also rewarding to develop new techniques to other backbone e.g. Transformer and GNNs based on our framework, which we leave for immediate future work. We hope that this work could inspire future explanations and defense algorithms of adversarial attacks. Due to the nature of this work, there may not be any potential negative social impact that is easily predictable.

# Acknowledgement

This work was partly supported by National Key Research and Development Program of China (2020AAA0107600), National Natural Science Foundation of China (61972250, 72061127003), and Shanghai Municipal Science and Technology (Major) Project (2021SHZDZX0102, 22511105100).

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
