In this Supplementary Material, we first present details of the Shapley value sampling (Appendix A). Then we give more experimental results on CIFAR-100 and stability analysis of Shapley value (Appendix B). We further show the training details and ablation study for the sample complexity of our CSA (Appendix C). Finally, we add properties of the Shapley value and proof of decomposition of CNNs in frequency domain (Appendix D).

## A Details of the Shapley Value Sampling

In this section, we introduce the details of the Shapley value sampling.

### A.1 Details of the Model for the Shapley Value Sampling

We sample the Shapley value for models trained on CIFAR10, CIFAR100 and ImageNet. For CIFAR10 and CIFAR100, we employ ResNet-18 and train them ourselves. The ST model on CIFAR10 and CIFAR100 is trained for 100 epochs with SGD optimizer where learning rate is set to be 0.1 and decayed by 0.1 at 75-th epoch and 90-epoch. The momentum is set to be 0.9 and weight decay is set to be $2e - 4$. The AT model on CIFAR10 and CIFAR100 is trained with adversarial examples generated with PGD-10 [24] with a L-inf bound where epsilon is set to be $8/255$ and stepsize is set to be $2/255$. The other settings of the training of AT model are the same of the ST model. For ImageNet, we employ standard ResNet-50 provided in robustbench [11].

### A.2 Details of the Sampling of the Shapley Value

We employ Monte-Carlo sampling method for the Shapley value [8] to sample the Shapley value. For CIFAR10 and CIFAR100, we divide the image into $16 \times 16$ patches in the frequency domain where each patch contains $2 \times 2$ frequency components and sample 2000 times for each data sample. For ImageNet, we divide the image into $32 \times 32$ patches in the frequency domain where each patch contains $7 \times 7$ frequency components and sample 1000 times for each data sample.

### A.3 Analysis about the Error Bound of our Shapley Value Sampling

As discussed in previous work [26], once we know the variance of the marginal contributions we have an bound for the estimation error of the Shapley Value. For our estimated Shapley value $\hat{\phi}$, assume the probability of the estimation error being greater than $\epsilon$ is less than $\delta$ that is

$$Pr(\|\hat{\phi} - \phi\| \geq \epsilon) \leq \delta \qquad (15)$$

With Chebyshev's inequality, for marginal contributions with variance as $\sigma^2$, the required sample times should be $m \geq \lceil \frac{\sigma^2}{\delta \epsilon^2} \rceil$

We calculate the average variance of the marginal contribution of each patch of the images in the first 200 images of each class of the train set of CIFAR10. The average variance is $1.814 \times 10^{-3}$. With $\delta = 0.05$ and $\epsilon = 0.005$ the required sample times is around 1450. Therefore, we set the sample times on CIFAR10 and CIFAR100 as 2000. Limited by the calculation complexity, we set the sampling times for ImageNet at 1000.

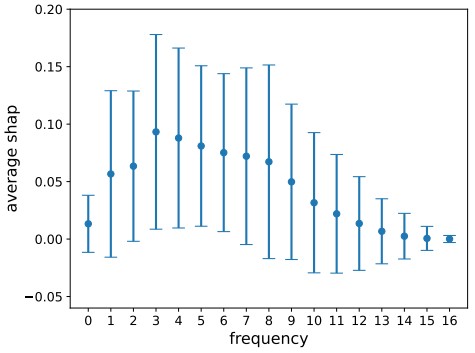

Figure 8: Average Shapley value where the length of errorbars indicate the standard deviation between different instances which demonstrate the instance disparity.

### A.4 Shapley Value Quantification Results

We demonstrate more Shapley Value Quantification Results on ImageNet and TinyImageNet in Fig. 9 and Fig. 10. Each row contains results of a data sample. We display the origin image, image reconstructed with PFCs, image reconstructed with NFCs and the heatmap of Shapley value from left to right. Note that the for the heatmap of Shapley value, red represents positive and blue

represents negative. The closer to the center the lower the frequency. To further demonstrate the instance disparity, we present the average Shapley value with the standard deviation between different instances in Fig. 8 which shows a large standard deviation.

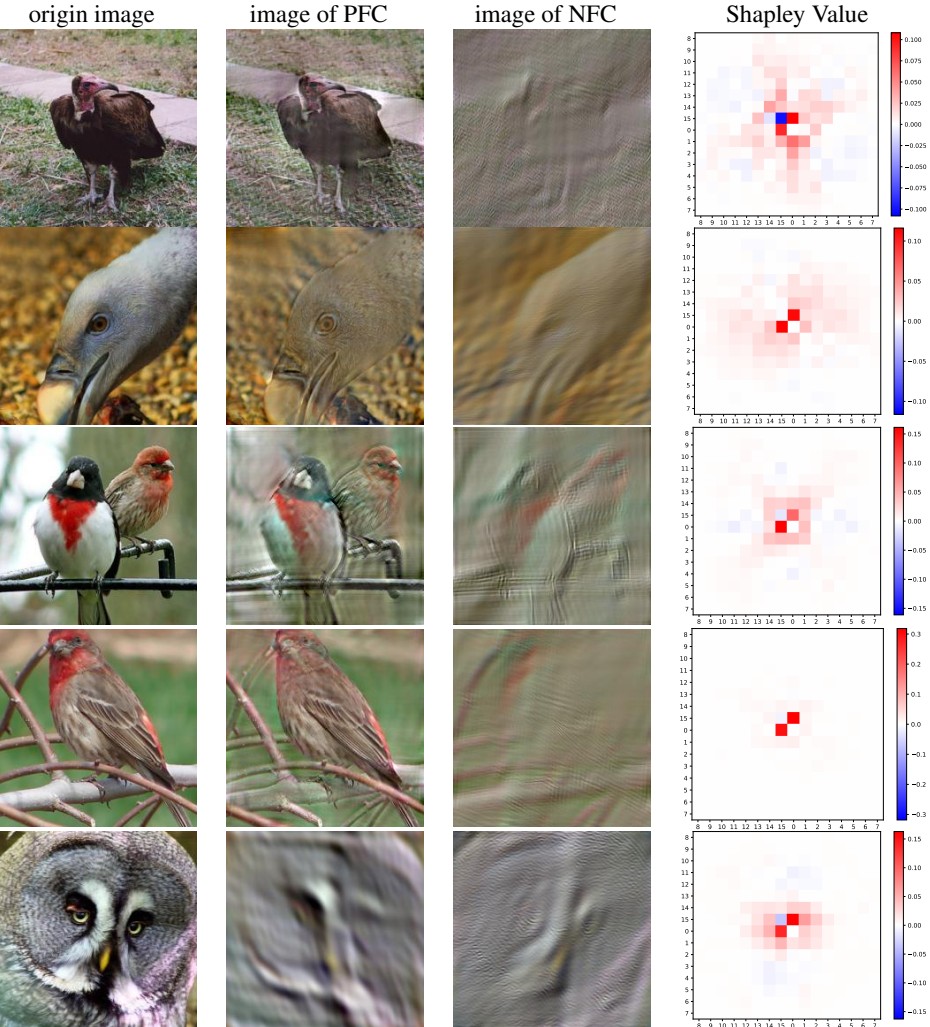

Figure 9: Shapley value of frequency components of data samples from ImageNet

## A.5 Time Complexity Analysis for The Sampling of the Shapley Value

The main cost of the sampling of the Shapley value on CNNs comes from generating masks and performing model inference. Assume that frequency components are devided in to $d \times d$ patches, for the size $d$, the time complexity is $\mathcal{O}(N^2)$. For sampling times, the time complexity is $\mathcal{O}(N)$.

We report the time consumed for sampling Shapley value for ResNet-18 on CIFAR10 with Intel(R) Core(TM) i7-7820X CPU @ 3.60GHz and one GeForce RTX 2080 Ti in Tab. 4.

# B Extra Experiment Results

## B.1 Experiment Results of vgg16 on CIFAR10

In this section we present the experiment results of vgg16 on CIFAR10, the results are similar to the results of ResNet-18. The dimension of fully connected layers of vgg16 is changed from $4096$ to $512$ in order to suit the scale of the image in CIFAR10. Fig. 11 shows the similar difference

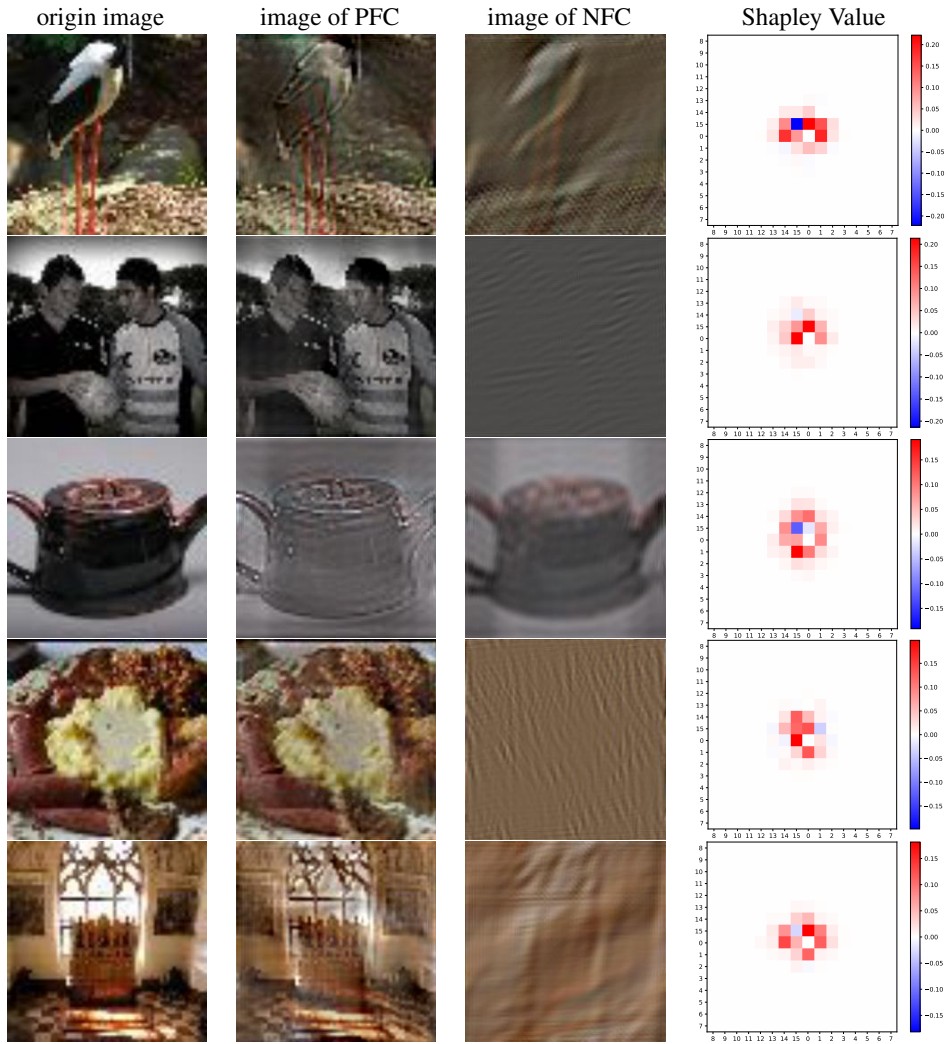

Figure 10: Shapley value of frequency components of data samples from TinyImageNet

| sampling times | of patches | time consumed(s) |
|---|---|---|
| 1000 | $16 \times 16$ | $79.58 \pm 5.21$ |
| 1000 | $32 \times 32$ | $357.37 \pm 7.49$ |
| 2000 | $16 \times 16$ | $151.50 \pm 5.34$ |
| 2000 | $32 \times 32$ | $630.67 \pm 13.64$ |

Table 4: Time consumed for sampling Shapley value for ResNet-18 on CIFAR10 with Intel(R) Core(TM) i7-7820X CPU @ 3.60GHz and GeForce RTX 2080 Ti. Each result is averaged over 10 test.

between adversarial trained vgg16 and standard trained vgg16. Fig. 12 demonstrate a similar negative relationship between the adversarial accuracy of AT model and the average absolute Shapley value of HFC for ST model with vgg16. Fig. 13 demonstrate the same attack bias for vgg16 as ResNet18.

## B.2 Experiment Results on CIFAR100

In this section, we present the experiment results on CIFAR100.

**Shapley value in frequency domain** For frequency domain analysis, we conduct the same experiments on CIFAR100 as on CIFAR10. As shown in Fig. 14, we present the average Shapley value

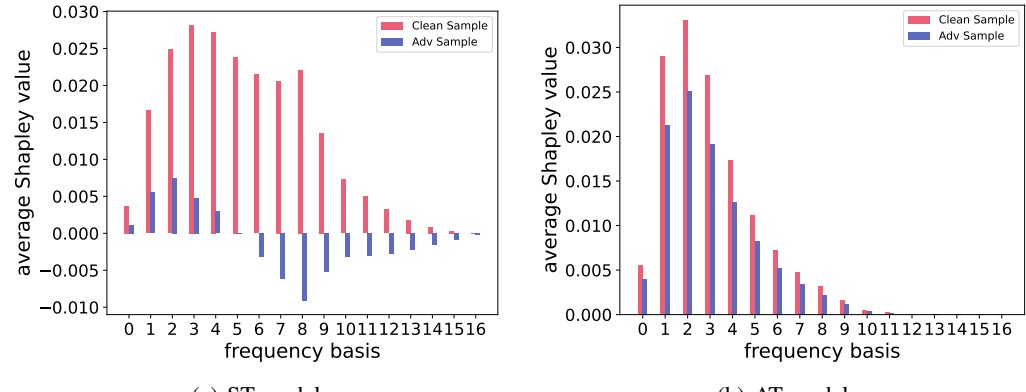

(a) ST model            (b) AT model

Figure 11: Average Shapley value of frequency componets in clean samples and adversarial samples of vgg16 over frequency basis on CIFAR10

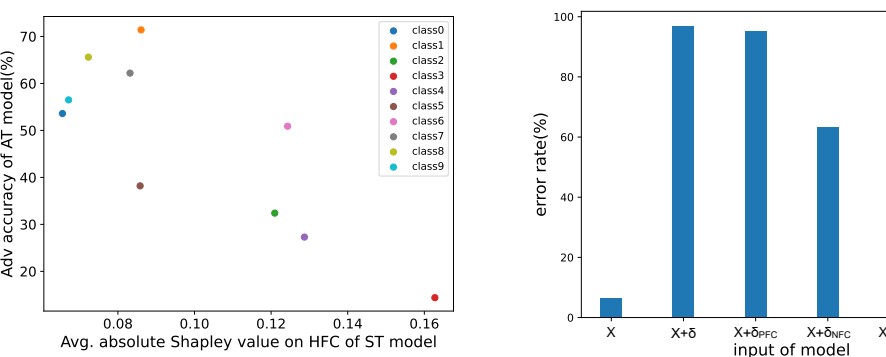

Figure 12: The negative relationship between the adversarial accuracy of AT model and the average absolute Shapley value of HFC for ST model(Experiment conducted on vgg16)

Figure 13: The error rate of vgg16 on CIFAR10 with different input. $X$ represent origin data. $\delta$ is the adversarial noise generated against the model.

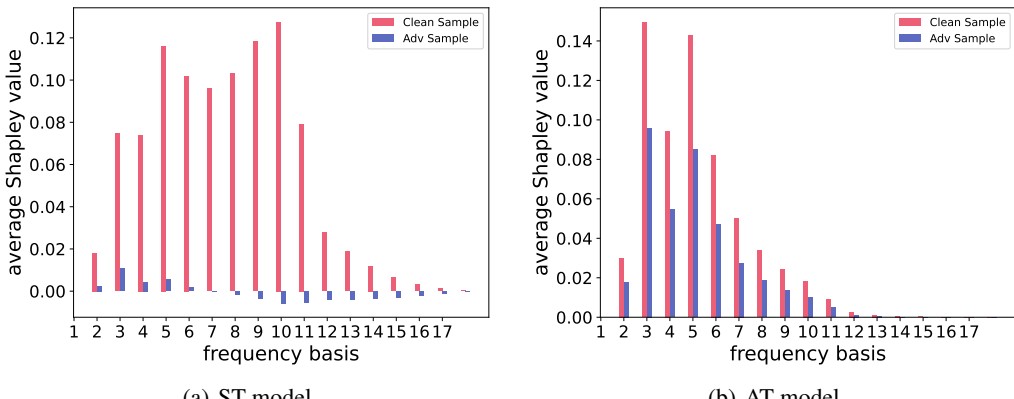

(a) ST model            (b) AT model

Figure 14: Average Shapley value of frequency componets in clean samples and adversarial samples of ResNet-18 over frequency basis on CIFAR100

of frequency components of the first 10 images of each class over frequency basis. The results are similar to the results on CIFAR10 except that the absolute average Shapley value of adversarial examples on ST model are relatively smaller.

### B.3  Stability analysis of Shapley Value

We show the variance of Average Shapley Value between different data samples to measure the statistical stability on Fig. 15. We plot the error bars of the average Shapley value of frequency components of clean samples on ST model and AT model on CIFAR10 and CIFAR100. The variance is relatively big on LFCs and small on HFCs. For HFCs, the average Shapley value and the variance on AT model is smaller than that on ST model and near zero which shows that AT model are less impacted by HFCs.

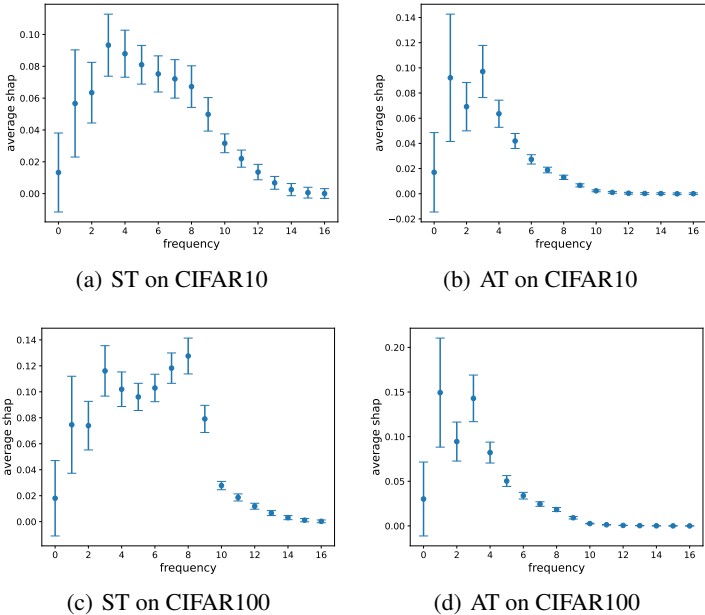

(a)  ST on CIFAR10          (b)  AT on CIFAR10

(c)  ST on CIFAR100         (d)  AT on CIFAR100

Figure 15: Average Shapley value of clean samples with error bars on CIFAR10 and CIFAR100. The length of the error bar is $2\sigma$ where $\sigma$ is standard error over 5 times of quantification.

## C  Details of Our Proposed Data Augmentation

### C.1  Training Details of CSA

As introduced in Eq. 13, we extract the CSA of images in the train set of CIFAR10 and CIFAR100 for ST model. We train model in the same manner as the AT model trained in Appendix A. Models are trained for 100 epochs with SGD optimizer where learning rate is set to be 0.1 and decayed by 0.1 at 75-th epoch and 90-epoch. The momentum is set to be 0.9 and weight decay is set to be $2e-4$. For other data augmentations on train data, we employ random horizontal flip with the probability set to 0.5. The adversarial examples are generated with PGD-10 [24] with a L-inf bound where epsilon is set to be $8/255$ and stepsize is set to be $2/255$. For TRADES[49], the trade-off regularization parameter $\beta$ is set to 6. The $\alpha$ in Eq. 14 linearly increase from 0 to 0.5 through the training process.

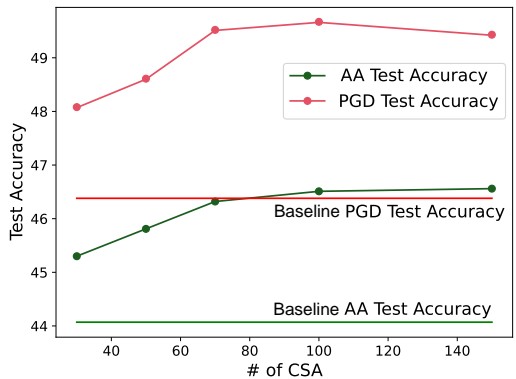

Figure 16: PGD and AA Test Accuracy of ResNet-18 trained with different number of CSA for each class on CIFAR10 at the last epoch

## C.2 Ablation study for the size of our CSA

The number of CSA we used for data augmentation may affect the effectiveness of the data augmentation. We generate CSA for the first 150 images of each class of CIFAR10 that is 3% of whole train set. We train ResNet-18 as described in Appendix C.1 with different number of CSA and test the model with AA [12] and PGD-20 [24] at the last epoch. The result is shown in Fig. 16.

As the number of CSA for each class on CIFAR10 increasing, the test accuracy under AA and PGD-20 attack gradually increase. At the same time the data augmentation with CSA is already effective with very few CSA *i.e.* 30 CSA per class which means the CSA we generate could represent features that are not correctly learned by the model.

## D Justification for the Shapley Value on CNNs in Frequency domain

### D.1 Properties of the Shapley value

As proved in the seminal work [31], the Shapley value is the only solution that satisfies the following desirable properties:

- **Null player**: A player $i$ is a null player if $\forall S \subseteq \mathcal{N}, V(S \cup \{i\}) = V(S)$. The Shapley value of every null player is zero.
- **Efficiency**: Shapley value satisfies that $\sum_{i \in \mathcal{N}} \phi_i^{\mathcal{N},V} = V(\mathcal{N})$
- **Symmetry**: Two players $i, j \in \mathcal{N}$ are symmetric if $\forall S \subseteq \mathcal{N} \backslash \{i, j\}, V(S \cup \{i\}) = V(S \cup \{j\})$. The Shapley value satisfies that for all symmetric players $i, j$, $\phi_i^{\mathcal{N},V} = \phi_j^{\mathcal{N},V}$
- **Linearity**: For two games $(\mathcal{N}, U), (\mathcal{N}, V)$, if a third game is defined as $(\mathcal{N}, W)$ where $\forall S \subseteq \mathcal{N}, W(S) = U(S) + V(S)$. It holds that $\phi_i^{\mathcal{N},W} = \phi_i^{\mathcal{N},U} + \phi_i^{\mathcal{N},V}$

### D.2 The Decomposition of Convolution in Frequency Domain

As shown in Remark 1, the output of a convolution operation can be decomposed into sum of output of frequency components of $X$. For a convolution kernel $C \in \mathbb{R}^{d_3 \times d_4}$, the element at $i$-th row and $j$-th column of the output of convolution function with the kernel $f_C$ is

$$f_C(X)(i,j) = \sum_{m=0}^{d_3-1} \sum_{n=0}^{d_4-1} X(i+m, j+n)C(m,n) \tag{16}$$

$$= \sum_{m=0}^{d_3-1} \sum_{n=0}^{d_4-1} \left[ \frac{1}{d_1 d_2} \sum_{u=0}^{d_1-1} \sum_{v=0}^{d_2-1} \mathcal{F}(X)(u,v) e^{i2\pi(\frac{(i+m)u}{d_1} + \frac{(j+n)v}{d_2})} \right] C(m,n) \tag{17}$$

$$= \sum_{u=0}^{d_1-1} \sum_{v=0}^{d_2-1} \mathcal{F}(X)(u,v) \sum_{m=0}^{d_3-1} \sum_{n=0}^{d_4-1} \frac{1}{d_1 d_2} e^{i2\pi(\frac{(i+m)u}{d_1} + \frac{(j+n)v}{d_2})} C(m,n) \tag{18}$$

We represent the $d_1 \times d_2$ Notch pass filter where all the elements equal to zero except the element on $u$-th row and $v$-th column is one with $I_{d_1 \times d_2}^{u,v}$. We have

$$\mathcal{F}^{-1}[I_{d_1 \times d_2}^{u,v}](i+m, j+n) = \frac{1}{d_1 d_2} e^{i2\pi(\frac{(i+m)u}{d_1} + \frac{(j+n)v}{d_2})} \tag{19}$$

Then we get

$$f_C(X)(i,j) = \sum_{u=0}^{d_1-1} \sum_{v=0}^{d_2-1} \mathcal{F}(X)(u,v) \sum_{m=0}^{d_3-1} \sum_{n=0}^{d_4-1} \mathcal{F}^{-1}[I_{d_1 \times d_2}^{u,v}](i+m, j+n)C(m,n) \tag{20}$$

$$= \sum_{u=0}^{d_1-1} \sum_{v=0}^{d_2-1} \mathcal{F}(X)(u,v) f_C \left[ \mathcal{F}^{-1}[I_{d_1 \times d_2}^{u,v}] \right](i,j) \tag{21}$$

Therefore, we get

$$f_C(X) = \sum_{u=0}^{d_1-1} \sum_{v=0}^{d_2-1} \mathcal{F}(X)(u,v) f_C \left[ \mathcal{F}^{-1}[I_{d_1 \times d_2}^{u,v}] \right] \tag{22}$$