# OpenReview forum: "Rethinking and Improving Robustness of Convolutional Neural Networks: a Shapley Value-based Approach in Frequency Domain"
_NeurIPS.cc/2022/Conference — NeurIPS 2022 Accept_

### Official Review · Reviewer_w1Z4 · 2022-07-05

**Rating:** 8
**Confidence:** 4
**Soundness:** 3 good
**Presentation:** 3 good
**Contribution:** 3 good

**Summary:**

This work provides an interesting perspective to the robustness of CNNs in frequency domain with the help of Shapley value via a theoretically stable quantification of the contribution of frequency components.

Specifically, the authors bring a new hypothesis about the trade-off between accuracy and robustness on adversarial training and an in-depth analysis of attack bias.

The authors also investigate the fairness problem in adversarial training and propose to boost robustness by proactively augmenting inputs with negative frequency components of their class. Experimental results verify the effectiveness of their method.

**Questions:**

Questions:

1. [Major concern] On line 34, the authors talk about a series of works improve robustness by filtering out HFC, while these baselines are missing in their experiment section.

2. [Minor concern] On line 261-265, the authors claim that though calculating Shapley value requires large sampling steps, their strategy only depends on a small ratio of data. It may be better to provide a detailed time complexity of this overhead.


**Limitations:**

Yes, the authors addressed the limitations.

**Strengths And Weaknesses:**

Pros:
+ This work performs a novel fine-grained analysis of the contribution of every frequency component in an instance-level with Shapley value, while previous works only do the coarse division of frequency spectrum at dataset-level.
+ Based on the quantization of Shapley value, the authors propose novel and interesting explanations about the existence of adversarial examples and the success and limitations of adversarial training on CNNs.
+ This work also delves into the class-wise robustness and proposes novel explanations, which may shed light on mitigating the fairness problem on adversarial training. Their proposed data augmentation defense strategy gets evaluated under strong attack baselines that verify its effectiveness.

Cons:
- Lack of comparison with baselines which imporve robustness by filtering out HFC.
- Lack of time complexity analysis.

---

> ### Author Response · Authors · 2022-08-02
> **Response to Reviewer w1Z4**
>
> We sincerely appreciate your recognition of the novelty of our method. We set out below our responses to each of the questions.
>
> **Q1: Lack of comparison with baselines which imporve robustness by filtering out HFC.**
>
> - **R1:** Most high frequency filtering methods are proposed to deal with $l_2$ norm adversarial attacks and are compared with other non-adversarial training defense methods[1][2]. We compare our method with the method proposed in [3] which combines filtering out HFCs with TRADES. We add the result in [Appendix  B.1]. The best result during training is also shown below:
>
>   | Method          | Clean Accuracy | Attacked by PGD-20 | Attacked by AutoAttack |
>   | --------------- | -------------- | ------------------ | ---------------------- |
>   | TRADES          | **81.23%**         | 51.18%             | 47.94%                 |
>   | TRADES + sup HFC | 78.14%      | 50.66%           | 45.47%              |
>   | TRADES + CSA                |    80.84%            |                **52.57%**    |              **48.94%**          |
>
>    For $l_\infty$ attack, suppressing high frequency components even slightly degrade the testing accuracy on CIFAR10.
>
> **Q2: Lack of time complexity analysis.**
>
> - **R2:** We conduct time complexity analysis and add the results to [Appendix A.5]. The main cost of the sampling of the Shapley value on CNNs comes from generating masks and performing model inference. Assume that frequency components are divided into $d\times d$ patches, for the size $d$, the time complexity is $\mathcal{O}(N^2)$. For sampling times, the time complexity is $\mathcal{O}(N)$.
>
>   We report the time consumed for sampling Shapley value for ResNet-18 on CIFAR10 with Intel(R) Core(TM) i7-7820X CPU @ 3.60GHz and one GeForce RTX 2080 Ti as below (each result is averaged over 10 test):
>
>   | Sampling times | # of patches  | time consumed (s)  |
>   | -------------- | ------------- | ------------------ |
>   | $1000$         | $16\times 16$ | $79.58 \pm 5.21$   |
>   | $1000$         | $32\times 32$ | $357.37 \pm 7.49$  |
>   | $2000$         | $16\times 16$ | $151.50 \pm 5.34$  |
>   | $2000$         | $32\times 32$ | $630.67 \pm 13.64$ |
>
> We wish the above response could relieve your current concern. We remain open to any of your further questions.

---

> > ### Comment · Reviewer_w1Z4 · 2022-08-08
> > **Response to rebuttal**
> >
> > Thank you to the authors for their response. I have read the authors' responses. My concerns have been addressed and I keep my initial rating.

---

> > > ### Author Response · Authors · 2022-08-08
> > > **Response to Reviewer w1z4**
> > >
> > > Thanks for your feedback and positive evaluation. We hope this work would provide a new perspective when it comes to explaining adversarial attack and adversarial training in frequency domain and have a broad impact.

---

### Official Review · Reviewer_mEHT · 2022-07-07

**Rating:** 6
**Confidence:** 4
**Soundness:** 3 good
**Presentation:** 2 fair
**Contribution:** 2 fair

**Summary:**

In this paper, the authors mainly focus on the problem of explaining the relationship of CNN's robustness and frequency bias by proposing a new method utilizing the Shapley-value as the major tool. They assign the Shapley value to the frequency domain components and futher examine its relationship with HFC and LFC. The main contributions are:
1. Propose a new method to explain the robustness--frequency bias relationship;
2. Conduct instance-level, class-level and dataset-level test on the frequency-component explanation experiment, reveal the relationship between HFC/LFC and PFC/NFC;
3. Propose a data augmentation method based on the explanation to improve the model robustness.


**Questions:**

Please refer to the previous section.

**Limitations:**

The authors conclude the only limitation as limited to CNNs. This does not look like a reasonable limitation, since the main topic is discussing the frequency bias of CNN. CNN is more likely to suffer from such biases becuase of its locality. Moreover, the limitations that I can think of are:
1. Lack of motivatiuon of using such a framework;
2. Lack of theoretical analysis on robustness-PFC/HFC.

**Strengths And Weaknesses:**

The main strengths are:
1. The problem is of broad interest and the angle is somehow novel, since the are few previous works focusing on instance-level explanation for CNN frequency bias;
2. The overall method is reasonable. Assigning Shapley value to the frequency domain components is also novel;
3. According to the results reported in the paper, the augmentation method also succeed in improving the model robustness.

The main weaknesses are:
1. The motivation of using Shapley value is not clear. In fact, analyzing the importance of each indivisual frequency components has been discussed before[1, 2]. The most intuitive explanation method would be simply replace the target frequency component with zero or mean value, and the results reported in the papers also align with each other. This also lead to another problem that this paper lacks baseline method to compare with, considering this is not a novel task;
2. Some explanation does not sound reasonable. For example, in section 4.5 the authors try to explain class-wise fairness with the Shapley values. However, the authors state that class 0 is both robust HFC-negative because the label of which is "transportation". This obviously make the previous robustness-HFC explanation less strong;
3. Lack of more experimental results. The authors argue that the HFC/LFC dependency is inconsistent across instances within the same dataset by giving the Shapley heatmap of a shark and a gold fish. It would be great if the authors could show more results, and also avergae results across the class/dataset to better illustrate, considering the "shark" image is kind of blurred and even hard to tell by humans;
4. Minor issues. For Figure 3, please double check the x-axis. For line 168, there is a typo with "LHC". For line 248, did you mean four classes? For Figure4, what do you mean by model output?


[1] Visualizing and Understanding Convolutional Networks, Zeiler et al;

[2] Towards Frequency-Based Explanation for Robust CNN, Wang er al.

---

> ### Author Response · Authors · 2022-08-02
> **Response to Reviewer mEHT[updated]**
>
> Thank you for your time and valuable comments. Our responses are as follows:
>
> **Q1: The motivation of using Shapley value is not clear. In fact, analyzing the importance of each indivisual frequency components has been discussed before[1, 2]. The most intuitive explanation method would be simply replace the target frequency component with zero or mean value, and the results reported in the papers also align with each other. This also lead to another problem that this paper lacks baseline method to compare with, considering this is not a novel task;**
>
> - **R1:** Thank you for your kind comments. We would like to show both the methodological advantage and new experimental findings of introducing Shapley value into the frequency domain compared with [1][2].
>
>   First, [1] visualizes convolutional filters of CNNs on the frequency domain, while we focus on how CNNs perceive inputs on the frequency domain. Moreover, [1] only shows qualitative results about HFC and LFC, without performing quantitative analysis.
>
>   Second, the attribution method proposed by [2] can be covered by our conceptual framework and is a biased estimate of Shapley value. The essence of Shapley value is the average expected difference brought by replacing the target frequency component under all possible circumstances where other frequency components are either replaced or not. Specifically, the method in [2] proposes to calculate the difference under one special circumstance where all the other frequency components are not replaced, which can be viewed as a biased estimation of the Shapley value. Moreover, in Appendix D, we show that the Shapley value enjoys several desirable theoretical properties in cooperative game theory[3], which the method in [2] does not satisfy.
>
>   As for experimental findings, [2], like other previous works[4][5], hold the hypothesis that the vulnerability of CNNs is due to reliance on HFC, while we find that the frequency bias of CNNs varies across samples (Fig. 2) and both the positive and negative frequency components exist across the whole frequency spectrum, in LFC and HFC (Fig. 7) which provide a possible explanation for the adversarial attack on LFCs[line234-238]. Based on our findings, we propose to proactively inject class-wise negative frequency components into each sample of the same class to enhance the robustness of CNNs (Sec. 5). Our strategy also outperforms baselines that simply utilize HFC filtering by a huge margin (Appendix B.1).
>
> **Q2: Some explanation does not sound reasonable. For example, in section 4.5 the authors try to explain class-wise fairness with the Shapley values. However, the authors state that class 0 is both robust HFC-negative because the label of which is "transportation".**
>
> - **R2:** We average the absolute Shapley value of HFCs of each class for ST model, and we find a strong negative correlation between values of HFCs on ST model and class-wise robustness on AT model [Fig. 5]. It supports the explanation that classes, where HFCs have larger impact on the ST model, would have relatively lower robust accuracy under AT. For class 0, we conjucture that because the other three class of transportation is robust it is harder for model to misclassify class 0.
>
> **Q3: Lack of more experimental results. The authors argue that the HFC/LFC dependency is inconsistent across instances within the same dataset by giving the Shapley heatmap of a shark and a gold fish. It would be great if the authors could show more results, and also avergae results across the class/dataset to better illustrate, considering the "shark" image is kind of blurred;**
>
> - **R3:** We have added more experimental results of quantification on ImageNet and TinyImageNet in [Appendix A.4] which further shows the instance disparity. We appreciate your suggestion and will add the average results to better illustrate in the future update.
>
> **Q4: Minor issues. For Figure 3, please double check the x-axis. For line 168, there is a typo with "LHC". For line 248, did you mean four classes? For Figure4, what do you mean by model output?**
>
> - **R4:** We have checked and revised these minor issues for Fig.3 and line 168. For line 248, we meant for the three classes 1,8,9 in "transprotation". For Fig.4, we demonstrate error rate instead in Fig.6 after revision. Thanks for pointing these out.
>
> We wish the above response could resolve your concerns and we will be truly grateful if you could reconsider the methodological novelty and significance of our work to the XAI community. We remain available for any further questions you may have.
>
> **Reference**
>
> [1] Visualizing and Understanding Convolutional Networks, Zeiler et al;
>
> [2] Towards Frequency-Based Explanation for Robust CNN, Wang er al;
>
> [3] A value for n-person games. Shapley;
>
> [4] High-frequency component helps explain the generalization of convolutional neural networks. Wang et al;
>
> [5] A fourier perspective on model robustness in computer vision. Yin et al.

---

> > ### Comment · Reviewer_mEHT · 2022-08-09
> > **Thank you for the clarification**
> >
> > I thank the authors for the clarification. Most of my concerns get addressed after the additional experiments provided. However, I still have a concern regrading the motivation, and also the relationship between PFC/NFC and LFC/HFC, since in the additional examples of heatmaps distribution (Fig 8, 9), it turns out that most of the PFC are located in the low-frequency area, in contrast to the original "shark" example. I am raising my rating to 5.

---

> > > ### Author Response · Authors · 2022-08-10
> > > **Further Response to Reviewer mEHT**
> > >
> > > Dear Reviewer mEHT,
> > >
> > > We sincerely thank you for your positive feedback and for updating the score. We further address your concerns regarding the motivation and the additional examples(Fig. 8, 9).
> > >
> > > - **The Motivation of our framework**
> > >
> > >     To inspect CNNs in frequency domain in a more fine-grained way, we evaluate the contribution of each frequency component for more thorough analyses. Therefore we introduce a more theoretically grounded and more sophisticated metric, Shapley value, to the frequency domain for our attribution method. Although [1] seeks the individual analysis of importance of each frequency component, as we pointed out in our response to Q1, the attribution method in [1] is actually a biased estimate of Shapley value.
> > >
> > >     We did base on Shapley value to discover novel findings and hyphotheses different from previous works including [1], like new hyphotheses about the fairness problem of AT (Sec. 4.3) and the attack bias on frequency domain (Sec. 4.4). Based on the assumption that negative frequency components(NFCs) compose features that are misaligned with other classes by the model, it further motivates us to propose the data augmentation method CSA to align the distribution of features from the same class and thus improve the model robustness during AT. Our proposed Shapley value-guided data augmentation strategy greatly improved robustness of CNNs (Table 1), which further demonstrated the soundness and efficacy of our Shapley value tool.
> > >
> > > - **Additional examples in Fig. 8, 9**
> > >
> > >     As you pointed out, we'd like to further interpret the quantification results in Fig.8 and Fig. 9, which indeed demonstrated the disparity in the HFC/LFC dependence among different samples. For example, the 2nd row in Fig.8 and the 1st and the 3rd row in Fig.9 demonstrate similar results to the original 'Shark' image (Fig. 2). For these data samples, HFCs are mostly PFC therefore PFCs compose the contour and texture of the object while NFCs contain LFC therefore NFCs compose the mainbody of the object. Other images in Fig. 8,9 are more like the original 'gold fish' image (Fig. 2) while there is no two images identical to each other, which indeed demonstrated the disparity in the HFC/LFC dependence among different samples.
> > >
> > >     For the relationship between HFC/LFC and PFC/NFC, we discuss it in Sec.4.4.2 and demonstrate that PFCs and NFCs coexist over the frequency spectrum (Fig. 7) even though the Shapley value of LFCs(PFCs) are significantly larger than that of HFCs(NFCs).
> > >
> > > We hope that our clarification above could address your concerns. Thanks again for your valuable time and suggestions.
> > >
> > > [1] Towards Frequency-Based Explanation for Robust CNN, Wang et al;

---

> ### Author Response · Authors · 2022-08-08
> **To Reviewer mEHT: Looking forward to your response.**
>
> Dear Reviewer mEHT:
>
> Does our response address your concerns? Your feedback would be really appreciated and will definitely help reaching an informative decision on our submission.
>
> Thank you!
>
> Best regards.
>
> Paper5594 Authors

---

### Official Review · Reviewer_tXAo · 2022-07-09

**Rating:** 4
**Confidence:** 3
**Soundness:** 3 good
**Presentation:** 2 fair
**Contribution:** 2 fair

**Summary:**

This paper presents a method for using Shapley values to estimate the importance of different frequency components of input images for the predictions of a CNN. In addition they propose a new method to train more robust classifiers by reducing the effect of frequency components that usually reduce the classification chance of target class.

**Questions:**

Would like clarification on the confusing sentences disucssed in weaknesses, as well as any arguments for why claims I had issue with are indeed well supported by data.

**Limitations:**

Only one sentence discussing limitations and no discussion regarding potential negative social impact. Would like to see at least some effort made to discuss potential impact, or at least acknowledge the fact that due to the nature of the work there may not be any that are easily predictable.

In addition I had some issues with the Checklist which was filled incorrectly. Specifically, authors state they have included code and data but this is not the case. In additions authors claim to have reported compute times hardware used which is also false. They also included a theoretical result but marked the corresponding sections in the checklist as N/A.


### Edit:
Thank you for the detailed response. I think the updated submission is noticeable stronger, and has largely addressed my concerns regarding soundness of the results, however I still have some concerns regarding the contributions and clarity of this submission. In response I have updated my review as follows:

Soundness: 2->3, Presentation 1->2, Rating 3->4.
In addition I have updated my confidence from 4->3 as my remaining concerns are more subjective and I feel less confident about them.

**Strengths And Weaknesses:**

Strengths:
- Relatively original combination of feature importance tools such as Shapley values and the analysis of adversarial examples in the frequency domain.
- Some interesting analysis, such as the average Shapley values of different frequencies in Figure 3, and analysis of accuracy in different classes in section 4.5.
- Related work and background explained quite clearly.

Weaknesses:

Most claims not sufficiently backed by evidence:
- Most claims are based on one experiment with one network and with margins that do not look significant to me. Some examples:
Line 186-188: It seems like this should be the case based on definitions alone but experimental data does not support it very well, with very small difference between PFC and NFC.  This to me suggests there may be issues with the methodology.
- Fig 4. also forms the basis for one of the main claims of the paper claiming that positive parts of HFC and negative parts of LFC are most easily attacked which I don't think is displayed in the figure at all.
- Analysis of Figure 7 not well supported by data. While there may be some trend towards "classes that rely on HFC would have relatively low accuracy and robustness under AT", it is not clear if this is the case and I would like to see a more quantitave measure or a simpler graph focused on this.
- Experimental results of Table 1 are not very convincing. Authors get a slightly more robust model than the baseline they have chosen at the cost of clean accuracy. Similar results could probably be achieved by changing some parameters of the baseline method. In addition performance is far below published state-of-the-art on RobustBench, even for methods using same ResNet-18 backbone.

Clarity:
- Overall the papers was very hard to follow. It has many typos and weird word choices especially in sections describing main findings of the paper which made it hard to understand what these findings are claimed to be. Examples:
- Line 56-57: “we find that ST models merely exploit HFC but are more vulnerable on HFC under adversarial attacks”, what does this sentence mean? The phrase “merely exploits” is used many times in the paper such as the title of section 4.2 but it is unclear to me what the authors mean by this.
- Line 169-171: “Thus we assume that the tension between accuracy and robustness possibly accounts for the overlooking of AT on HFC, especially positive ingredients.” What? Extremely confusing sentence.
- Line 181: We future split → we further split
- Line 208: “NFC take up a fairly certain ratio“, what does certain mean in this context?
- Figure 1 is hard to understand.
- Figure 7 should have same x-axis for all subfigures

Contributions:
I don't think the papers contributions are very significant. As explained above robust training results don't seem too useful, and while the paper claims to extend previous work based on dataset level statistics of different features into instance based analysis, I only found the dataset level analysis discussed in the paper useful. The main takeaway that I think is adequately supported by this paper is that $l_{\infty}$-norm adversarial attacks target high frequency features but this was already known from previous work.

---

> ### Author Response · Authors · 2022-08-02
> **Response to Reviewer tXAo (1/2) [updated]**
>
> Thanks a lot for your time and detailed feedback. Below we address each of your comments point-by-point.
>
> **Q1: Most claims are based on one experiment with one network and with margins that do not look significant to me. some examples: line 186-188.**
>
> - **R1:** Thanks for your careful reading. We conduct more experiments with VGG16 on CIFAR10 and with ResNet18 on CIFAR100. We get similar results that well support our claims. These results have been added to [Appendix B].
>
>     For the original line 186-188, We analyze the difference of PFC and NFC by inspecting model outputs. In the revised version, we show the test error rate of ST model under adversarial attacks restricted to different frequency components such as PFC or NFC. As Fig. 6 shows, the significant difference between PFC and NFC supports our claim that "Adversarial attack is more effective on PFCs than on NFCs and is more effective on HFCs than on LFCs"[line 218-222].
>
> **Q2: Fig 4. also forms the basis for one of the main claims of the paper claiming that positive parts of HFC and negative parts of LFC are most easily attacked which I don't think is displayed in the figure at all.**
>
> - **R2:** Thank you for pointing at this! During rebuttal, we do an in-depth analysis of the coupling effect among HFC, LFC, PFC, NFC. Specifically, we take adversarial examples as input, mask their HFCs and LFCs respectively, and further mask the positive or negative parts of these examples. The table below shows the test error rate of ST models under different situations, which validates our claim in Sec. 4.4.1 about the attack bias that "adversarial attacks on PFCs and HFCs are more effective". However, the results do NOT show that negative parts of LFC are more easily attacked. Therefore, we have removed this claim in our updated version.
> | Adversarial Noise Type                     | Error Rate |
> | ------------------------------------------ | ---------- |
> | Adversarial noise on HFC | 84.50%     |
> | Adversarial noise on LFC  | 24.23%     |
> | Adversarial noise on PFC   | 95.07%     |
> | Adversarial noise on positive parts of HFC | 45.23%     |
> | Adversarial noise on negative parts of HFC | 30.40%     |
> | Adversarial noise on positive parts of LFC | 12.17%     |
> | Adversarial noise on negative parts of LFC | 7.63%      |
>
>     We further highlight that "PFCs and NFCs actually co-exist on every frequency basis" (Sec. 4.4.2), which is supported by the visualization of the distribution of PFCs and NFCs over the whole frequency spectrum in Fig. 7 (Fig.5 before revision). It provides a possible reasonable explanation for adversarial attacks improved by operating on LFCs in previous works[5][6]: adversarial attacks succeed by suppressing PFCs or boosting NFCs no matter where these frequency parts reside in HFCs or LFCs (Sec. 4.4.2). At last, we hope that our findings and hypotheses about attack bias could bring more attention on the contribution of individual frequency component when it comes to the explainability of adversarial examples.
>
> **Q3: Analysis of Figure 7 not well supported by data. I would like to see a more quantitative measure or a simpler graph focused on this.**
>
> - **R3:** Following your suggestion, based on results in Fig. 7 (Fig. 4 after revision), we average the absolute Shapley value of HFCs of each class on ST model as a proxy of the impact of HFCs on ST model. Then we find a strong negative correlation between values of HFCs on ST model and robustness on AT model (the Pearson correlation coefficient is $-0.8764$), which makes our hypotheses more convincing: classes whose HFCs have a larger impact on ST model, would have relatively lower robustness under AT.
>
> **Q4: Experimental results of Table 1 are not very convincing. Authors get a slightly more robust model than the baseline they have chosen at the cost of clean accuracy. Similar results could probably be achieved by changing some parameters of the baseline method. In addition, performance is far below published state-of-the-art on RobustBench, even for methods using the same ResNet-18 backbone.**
>
> - **R4:** We present our experiment results with WideResNet-28-10 on CIFAR10 in [Tab.1(b)]. Our data augmentation strategy CSA integrated with PGD-AT and TRADES significantly improves robustness over baselines. It is worth mentioning that our CSA further obtains better clean accuracy combined with TRADES while the cost of clean accuracy of CSA with PGD-AT could be negligible.
> | Method       | Clean Accuracy | Attacked by PGD-20 | Attacked by AutoAttack |
> | ------------ | -------------- | ------------------ | ---------------------- |
> | PGD-AT       | **85.90%**         | 50.18%             | 47.86%                 |
> | PGD-AT + CSA | 85.48%         | **53.00%**             | **49.61%**                 |
> | TRADES       | 84.13%         | 53.17%             | 50.21%                  |
> | TRADES + CSA             |    **84.41%**           |    **54.32**                |          **51.40%**             |

---

> ### Author Response · Authors · 2022-08-02
> **Response to Reviewer tXAo (2/2) [updated]**
>
> **Q5: Clarity of Line 56-57: “we find that ST models merely exploit HFC but are more vulnerable on HFC under adversarial attacks”**
>
> - **R5:** We have revised it and it is a typo where we meant for "barely". The message we want to deliver is "Standard Trained Models Mainly Exploit LFC but Are More Vulnerable on HFC" [Sec.4.2.1].
>
> **Q6:  Clarity of Line 169-171: “Thus we assume that the tension between accuracy and robustness possibly accounts for the overlooking of AT on HFC, especially positive ingredients.”.**
>
> - **R6:** Here we give a possible explanation for a common limitation of AT, i.e., the trade-off between accuracy and robustness. On [Sec4.2.2 line 175-177] (Sec. 4.3 line 165-167 before revision), we find that adversarially trained models improve robustness by filtering HFC, and "as Fig. 3(a) shows, the HFC of clean images plays a remarkable positive impact on model prediction" [line 178] (line 168 before revision). So we conjecture that the degradation of accuracy with AT is probably due to the HFC filtering effect of AT. We have revised it for better clarity.
>
> **Q7: Clarity of Line 208: “NFC take up a fairly certain ratio“, what does certain mean in this context?**
>
> - **R7:** We have revised it as "take up a fair ratio(at least 0.1)"[line230]
>
> **Other issues regarding clarity, discussion of limitations, and the Checklist**
>
> - **Re:** We sincerely thank the reviewer for pointing out these issues. We have revised those according to your suggestion. We have revised Fig. 1 and Fig. 7 (Fig. 4 after revision). We also have acknowledged in conclusion [Sec. 7] that there may not be any negative social impact of our work that is easily predictable. For the issues with the Checklist, we have included our code in the supplementary material and performed time complexity analysis in [Appendix A.5].
>
> **Concern regarding the contribution of the work**
>
> - **Re:** We respectfully thank you again for your thoughtful comments! In our humble opinion, we would like to clarify our motivation for introducing Shapley value in the frequency domain and the contribution of our work.
>
>     Most previous works[1][2] simply divide the frequency domain into two parts, i.e., HFC and LFC, and qualitatively evaluate the contribution of each part on predictions of CNNs, overlooking the possible disparity in instance-level, and frequency component-level. With help of Shapley value, we are capable of quantitatively inspecting the frequency spectrum in a fine-grained way:
>
>    (i). We provide quantitative results of each instance and our instance-level analysis shows that the frequency bias of CNNs actually varies greatly among samples [Sec 4.1].
>
>    (ii) To compare with previous works, we investigate the dataset-level statistics and provide new insights into the accuracy-robustness trade-off brought by adversarial training [Sec. 4.2].
>
>
>    (iii). Our instance-level observations motivate us to study the fairness problem induced by the class-wise disparity via quantifying the frequency component-level contribution [Sec. 4.3].
>
>    (iv). Our frequency component-level analysis further shows that both positive and negative frequency components exist across the whole frequency spectrum, either HFC or LFC [Fig. 7], and we find that adversarial attacks mainly exploit positive frequency components for success [Fig. 6], which is different from the common hypothesis that adversarial attacks succeed by exploiting HFC [1][3][4].
>
>    (v). Following our analysis of class-wise disparity, to enhance the robustness of CNNs, we propose to inject negative frequency components of one class into each sample of the same class and show its effectiveness under strong attacks like AutoAttack on CIFAR10 and CIFAR100 [Sec. 5].
>
> We follow your valuable suggestions and add more experiment results during the rebuttal. Our findings and hypotheses are consistent across different datasets, model capacities, and model architectures (Appendix B). Therefore, we sincerely hope that you could reconsider the significance of our contributions. We remain available for any further questions you may have and we look forward to your feedback at your earliest convenience.
>
> **Reference**
>
> [1] High-frequency component helps explain the generalization of convolutional neural networks. Wang et al;
>
> [2] A fourier perspective on model robustness in computer vision. Yin et al;
>
> [3] Feature distillation: Dnn-oriented JPEG compression against adversarial examples. Liu et al;
>
> [4] SHIELD: fast, practical defense and vaccination for deep learning using JPEG compression. Das et al；
>
> [5] Low frequency adversarial perturbation. Guo et al;
>
> [6] On the effectiveness of low frequency perturbations. Sharma et al.

---

### Official Review · Reviewer_em14 · 2022-07-13

**Rating:** 7
**Confidence:** 4
**Soundness:** 3 good
**Presentation:** 3 good
**Contribution:** 4 excellent

**Summary:**

This work study the adversarial robustness of convolutional neural networks (CNNs) based on the popular hypothesis about the frequency bias phenomenon in CNNs. By introducing a Shapley value, they quantify the impacts of every frequency component at an instance-level. They also propose a Shapley-value guided method for adversarial training.

**Questions:**

Please address questions in Weaknesses.

**Limitations:**

No potential negative societal impact observed.

**Strengths And Weaknesses:**

Strengths:
1. Using the Shapley value to analysis frequency components in each instance is an interesting idea. Their experiments also demonstrate that this metric is meaningful to improve the adversarial robustness of CNNs.
2. Their observations in Section 4 lead to instructive and constructive arguments on adversarial robustness. For example, their explanations about standard trained models (4.2) and adversarial trained models (4.3) are inspiring.

Weaknesses:
1. The experiment in Table 1 only use a small ResNet-18 model. Experiments on larger models, e.g., WideResNet-32, can be more convincing than this one.
2. In Figure 7, only class 3 shows large negative values on high-frequency components, while others only show small negative values or all positive values. Is there any intuitive or theoretical explanation for this?

---

> ### Author Response · Authors · 2022-08-02
> **Response to Reviewer em14**
>
> We sincerely appreciate your encouraging evaluation and valuable feedback. In the following, we address your comments point-by-point.
>
> **Q1: The experiment in Table 1 only uses a small ResNet-18 model. Experiments on larger models, e.g., WideResNet-32, can be more convincing than this one.**
>
> - **R1:** We apply our data augmentation strategy CSA to a large model, WideResNet-28-10 [Table 1(b)] and experimental results of WideResNet-28-10 on CIFAR10 further demonstrate the effectiveness of our CSA, which boost both the clean accuracy and robustness of baselines. We also add more experiment results of vgg16 on CIFAR10 and ResNet18 on CIFAR100 in Appendix B. The experiment results of WideResNet-28-10 on CIFAR10 are also shown below:
> | Method       | Clean Accuracy | Attacked by PGD-20 | Attacked by AutoAttack |
> | ------------ | -------------- | ------------------ | ---------------------- |
> | PGD-AT       | **85.90%**         | 50.18%             | 47.86%                 |
> | PGD-AT + CSA | 85.48%         | **53.00%**             | **49.61%**                 |
> | TRADES       | 84.13%         | 53.17%             | 50.21%                  |
> | TRADES + CSA             |   **84.41%**            |    **54.32**                |          **51.40%**              |
>
>
>
> **Q2: In Figure 7, only class 3 shows large negative values on high-frequency components, while others only show small negative values or all positive values. Is there any intuitive or theoretical explanation for this?**
>
> - **R2:** It is an interesting question and our intuitive explanation is that, for class 3(cat) in CIFAR10, the small textures of cats that correspond to HFCs vary greatly among different samples which could pose a negative contribution to the classification of cats.
>
>     To further reveal the relationship between values of HFCs and class-wise robustness, based on the results in Fig. 4 (the Fig.7 before revision), we average the absolute Shapley value of HFCs of each class for ST model, and we find a strong negative correlation between values of HFCs on ST model and class-wise robustness on AT model [Fig. 5].
>
>     As Fig. 5 shows, class 3 has the largest Shapley value of HFCs for ST model and the lowest robustness, which further confirms our claim about fairness in AT that large Shapley values, especially negative ones, on HFCs of one class could greatly degrade its robustness.
>
> We wish the above response could relieve your current concern. We remain open to any of your further questions.

---

> ### Author Response · Authors · 2022-08-08
> **To Reviewer em14: Looking forward to your response.**
>
> Dear Reviewer em14:
>
> Does our response address your concerns? Your feedback would be really appreciated and will definitely help reaching an informative decision on our submission.
>
> Thank you!
>
> Best regards.
>
> Paper 5594 Authors

---

### Author Response · Authors · 2022-08-02
**Thank All Reviewers and the Following Revisions are Made**

Dear Area Chair and Reviewers,

We would like to express our sincere gratitude to all reviewers for their valuable comments. We have carefully checked and answered questions of each reviewer, reoriganized the presentation order of our paper for better clarity, and added more experimental results that further support our claims. Here we summarize our revisions made so far:

- **Optimize Presentation Order of Section 4**:

    We ather the original Sec. 4.2 and 4.3 into a new Sec. 4.2 as its subsections respectively, and move the original Sec. 4.4 as the last subsection of Section 4. In this way, Section 4 is presented in a more clear logical flow:

    1. We first perform instance-level analysis in Sec. 4.1.

    2. We then perform both dataset-level and class-level analysis for the success and limitations of adversarial training in Sec. 4.2 and Sec. 4.3.

    3. we finally perform the analysis for the attack bias on frequency domain in Sec. 4.4.

- **Add More Experimental Results to Support Our Claims**:

    1. **Fairness in Adversarial Training (Fig. 5)**: Based on the original results on Fig. 4, we obtain the average absolute Shapley value of HFCs of each class on ST model, and find a strong negative correlation between values of HFCs on ST model and robustness on AT model, which makes our hypotheses more convincing.
    2. **Attack bias on Frequency Domain (Fig. 6)** In our original paper, we analyze the disparity of ST model outputs induced by different adversarial attacks, which are restricted to different frequency components. On Fig. 6, we show the test error rate of the ST model under those attacks, which further demonstrate our claim about attack bias: "Adversarial attack is more effective on PFCs than on NFCs and is more effective on HFCs than on LFCs"[line 218-222]".
   3. **More qualitative and quantification results in [Appendix A.4]** for the instance disparity on frequency domain.

   4. **Extra Experiment results on VGG16** show the consistent tendency with ResNet-18 and further support our claims [Appendix B.2].

- **Add More Experimental Results of our data augmentation strategy CSA**:
    1. **Time complexity analysis** [Appendix A.5].

    2. **Experimental results with WideResNet-28-10 on CIFAR10** [Table 1(b)]:

       | Method       | Clean Accuracy | Attacked by PGD-20 | Attacked by AutoAttack |
       | ------------ | -------------- | ------------------ | ---------------------- |
       | PGD-AT       | **85.90%**         | 50.18%             | 47.86%                 |
       | PGD-AT + CSA | 85.48%         | **53.00%**             | **49.61%**                 |
       | TRADES       | 84.13%         | 53.17%             | 50.21%                  |
       | TRADES + CSA             |    **84.41%**           |    **54.32**                |          **51.40%**             |

       The above results show that our method CSA could **boost both the clean accuracy and robust accuracy** on a larger model WideResNet-28-10.

   3. **More frequency-based defense baselines:** We add a baseline method that simply suppresses HFCs on CIFAR10 [Appendix B.1], whose performance is worse than our CSA.

        | Method          | Clean Accuracy | Attacked by PGD-20 | Attacked by AutoAttack |
        | --------------- | -------------- | ------------------ | ---------------------- |
        | TRADES          | **81.23%**         | 51.18%             | 47.94%                 |
        | TRADES + sup HFC | 78.14%      | 50.66%           | 45.47%              |
        | TRADES + CSA                |    80.84%            |                **52.57%**    |              **48.94%**          |

        The above results show the best result during training. It shows that our proposed method outperforms the baselines that filter out HFCs.

The updated contents are highlighted by blue text in the revised paper.

---

### Author Response · Authors · 2022-08-08
**Inquiry for Post-Rebuttal Comments (8/8/22)**

Dear Reviewers,

We would like to express our sincere gratitude for your valuable comments on this paper. We would also like to thank Reviewer tXAo for responding to our rebuttal. We are glad that our response has largely addressed your concerns and we remain available for your further  questions and concerns.

As the discussion is approaching its ending, for other reviewers, we humbly look forward to hearing from you about whether our rebuttal has addressed your concerns. If you have any further questions and concerns, feel free to post comments so that we can respond to your questions and concerns.

Thank you very much!

With best regards,

Paper5594 Authors

---

### Meta-Review · Area_Chair_3LDS · 2022-09-02

**Recommendation:** Accept
**Confidence:** Less certain

**Metareview:**

All reviewers find the paper's approach on using Shapley values to measure the impact of frequency spectrum on the adversarial robustness of CNNs novel and the results interesting. Initially there were concerns about limited evaluation of the proposed approach and presentation. Author's response provided additional results and better presentation of the paper. This has largely alleviated reviewer's main concerns and most of them are positive about the work. One reviewer still rates the paper negatively and are not fully convinced about the effectiveness of the proposed approach. I think overall the response seems positive to me and I suggest acceptance. I encourage authors to update the paper incorporating the reviewers suggestions and adding more experimental results that came up during the discussion.

**Award:**

No

---

### Decision · Program_Chairs · 2022-09-14

Accept